# Two Modalities Are Better Than One:
# Efficient Adversarial Purification via Multimodal Diffusion Models

**Mingyuan Bai** [1]  **Wei Huang** [2,3]  **Tenghui Li** [4]  **Andong Wang** [1]  **Chao Li** [1]  **Cesar F Caiafa** [1,5]  **Junbin Gao** [6]
**Qibin Zhao** [1]

## Abstract

Adversarial purification uses generative models to restore clean data distributions from unseen attacks without retraining classifiers. However, unimodal diffusion-based approaches struggle to preserve semantic consistency, while recent multimodal variants rely on computationally expensive adversarial training or distillation. Both approaches often lack theoretical guarantees. In this work, we propose MultiDAP, a novel framework leveraging multimodal diffusion models for efficient adversarial purification. MultiDAP first learns continuous class-agnostic prompts from clean data to capture rich semantic priors, replacing rigid hand-crafted templates. Guided by these prompts, MultiDAP purifies adversarial inputs by minimizing a regularized DDPM loss for only a few steps (e.g., 5-20). We provide theoretical guarantees for both the likelihood improvement via prompt learning and the convergence of the purification process. Extensive experiments on CIFAR-10, CIFAR-100, and ImageNet-1K demonstrate that MultiDAP matches the robustness of state-of-the-art baselines but with improved efficiency.

## 1. Introduction

Adversarial defense is fundamentally concerned with recovering the true semantic label information from adversarial

[1]Tensor Learning Team, Center of Advanced Intelligence Project, RIKEN, Tokyo, 1030027, JAPAN [2]Deep Learning Theory Team, Center of Advanced Intelligence Project, RIKEN, Tokyo, 1030027, JAPAN [3]The Institute of Statistical Mathematics, Tachikawa, Tokyo, 1908562, JAPAN [4]School of Automation, Guangdong University of Technology, Guangzhou, Guangdong Province, 510006, CHINA [5]Instituto Argentino de Radioastronomìa, CONICET CCT La Plata/CIC-PBA/UNLP, V. Elisa, 1894, ARGENTINA [6]Discipline of Business Analytics, University of Sydney, Darlington, New South Wales, 2008, AUSTRALIA. Correspondence to: Qibin Zhao <qibin.zhao@riken.jp>.

*Proceedings of the $43^{rd}$ International Conference on Machine Learning*, Seoul, South Korea. PMLR 306, 2026. Copyright 2026 by the author(s).

examples which are perturbed by human imperceptible but carefully crafted noise for deep learning classifiers to predict incorrect labels (Goodfellow et al., 2014). Adversarial purification has recently emerged as a promising paradigm for adversarial defense (Shi et al., 2021; Yoon et al., 2021; Nie et al., 2022; Wang et al., 2022; Bai et al., 2024; Lei et al., 2025). Unlike adversarial training (Croce & Hein, 2020; Laidlaw et al., 2021; Dolatabadi et al., 2022; Wang et al., 2023), which explicitly trains classifiers on adversarial examples, adversarial purification methods employ generative models to remove adversarial perturbations before classification (Song et al., 2017; Nie et al., 2022). This strategy offers two key advantages. (1) It does not need to retrain classifiers on generated attacks, thereby reducing computational overhead. (2) It provides stronger generalization to unseen adversarial attacks, as adversarial purification directly restores clean data distributions. However, early adversarial purification methods are based on generative adversarial networks (GANs) and energy-based models (EBMs) and fall behind adversarial training methods, because of their limited generative power (Nie et al., 2022).

Diffusion models have rapidly become the mainstream approach for adversarial purification due to their remarkable generative power and ability to approximate complex data distributions (Nie et al., 2022; Wang et al., 2022; Chen et al., 2024b; Zhang et al., 2025; Bai et al., 2024). By progressively adding Gaussian noise and removing it, diffusion models can effectively reconstruct clean samples from corrupted or perturbed inputs, making them particularly well-suited for removing adversarial perturbations (Nie et al., 2022). However, most existing work relies on unimodal diffusion models which attempt to preserve semantic information implicitly by injecting Gaussian noise to a specific level in the forward process and then denoising the input. Hence unimodal approaches often struggle to fully obtain semantic label information, limiting defense against stronger or adaptive attacks.

To address these limitations, recent work has explored multimodal diffusion models for adversarial purification, e.g., one-step control purification (OSCP) which leverages ControlNet to incorporate additional conditioning signals such as textual prompts (Lei et al., 2025). This line of work

is motivated by the observation that semantic/contextual information can help maintain label consistency under perturbations, whereas pixel-space classifiers may be brittle under distribution shifts induced by adversarial attacks (Zhou et al., 2024). However, existing multimodal approaches still face critical challenges. First, many methods require costly adversarial training or heavy distillation to achieve robust cross-modal alignment, which increases preparation cost and reduces deployability with off-the-shelf backbones. Second, training-free alternatives still rely on iterative purification, resulting in high test-time latency and limiting real-time defense. Third, theoretical guarantees on multimodal purification effectiveness and convergence remain limited, hindering principled design and scalability.

In this paper, we propose Multimodal Diffusion for Adversarial Purification (MultiDAP), a framework that leverages a single text-to-image diffusion backbone. Unlike unimodal diffusion purification methods that rely only on image features, MultiDAP conditions the diffusion model on learned continuous prompts to inject semantic priors. We learn these class-agnostic prompts from clean large-scale text–image pairs, exploiting the rich contextual representations of the text encoder while keeping the diffusion backbone frozen. At test time, MultiDAP performs prompt-guided purification by optimizing a regularized, prompt-conditioned DDPM objective for only a few steps (e.g., 5–20), substantially reducing inference cost compared to long-chain diffusion purification (e.g., DiffPure). Experiments on CIFAR-10, CIFAR-100 and ImageNet-1K show that MultiDAP achieves competitive or improved zero-shot robustness while being significantly more efficient. These results highlight the dual contribution of our work: introducing a theoretically grounded framework for adversarial purification and delivering practical improvements for real-world deployment.

## 2. Related Work

**Unimodal Diffusion Models for Adversarial Defense.** Diffusion models have demonstrated remarkable performance in generative tasks, owing to their ability to progressively refine noisy data to high-quality output. This generative nature has been explored for robustness in various contexts, including adversarial purification (Nie et al., 2022), adversarial training (Wang et al., 2023) and robust classification methods (Chen et al., 2024b). A notable application of diffusion models lies in purification-based defenses, where adversarially perturbed inputs are restored to their clean counterparts. Methods leveraging guided diffusion models have shown efficacy in removing perturbations while preserving the underlying data features, making them suitable for tasks like classification (Lee & Kim, 2023; Xiao et al., 2022; Bai et al., 2024; Yeh et al., 2024). Additionally,

diffusion-based classifier have gained traction by integrating generative and discriminative modelling (Zimmermann et al., 2021; Clark & Jaini, 2023; Chen et al., 2024b). Their robustness to input perturbations and adversarial attack is attributed to optimal empirical score function (Chen et al., 2024a). While these approaches highlight the versatility of diffusion models in adversarial defense, they often face challenges in efficiency, effectiveness from unimodality and theoretical guarantees, motivating further investigations.

**Multimodal Approaches in Adversarial Defense.** Vision-language models (VLMs) suggest that auxiliary modalities can improve robustness by injecting semantic priors beyond pixel space. For example, CLIP (Radford et al., 2021) aligns images and text in a shared embedding space, which has inspired adversarial finetuning (Schlarmann et al., 2024), adversarial prompt tuning (Zhang et al., 2024; Li et al., 2024; Sheng et al., 2025), and other VLM-based defenses (Wang et al., 2025). Despite these advances, most multimodal robustness methods remain training-intensive, typically requiring adversarial training or distillation for robust cross-modal alignment. In multimodal diffusion purification, OSCP (Lei et al., 2025) improves semantic consistency via text control, but still incurs high cost without distillation and offers limited theoretical guarantees. These limitations motivate methods that combine multimodal semantic priors with efficient purification and provable guarantees, as pursued in our MultiDAP.

## 3. Multimodal Diffusion Models for Adversarial Purification

### 3.1. Problem Setup

Let $x \in \mathcal{X}$ denote a clean input with label $y$, and $x^{adv} = x + \delta$ be an adversarial example generated under perturbation constraint $\|\delta\|_p \leq \epsilon$. Here the perturbation $\delta$ is constrained under an $\ell_p$-norm threat model, with $p \in \{2, \infty\}$ being the most common cases. The $\ell_\infty$ attack bounds the maximum per-pixel distortion, ensuring imperceptibility, while the $\ell_2$ attack restricts the overall perturbation.

Adversarial purification aims to transform an adversarial input $x^{adv}$ back to a sample $x^{pur}$ that lies close to the clean data manifold, such that $f(x^{pur}) = y$. Recent works have demonstrated that diffusion models are particularly well suited for this task, due to their strong generative ability to approximate complex data distributions (Nie et al., 2022).

A diffusion model (Song et al., 2020) defines a forward noising process that gradually perturbs clean data $x_0$ into Gaussian noise through a sequence of latent variables $\{x_t\}_{t=0}^T$:

$$q(x_t \mid x_{t-1}) = \mathcal{N}\left(x_t; \sqrt{1 - \beta_t}\, x_{t-1}, \beta_t I\right),$$

where $\{\beta_t\}$ is a variance schedule (Ho et al., 2020). This process ensures that as $t \to T$, the sample $x_T$ approaches pure noise, as $T$ is large enough. The reverse denoising process is parameterized by a neural network, which predicts the added noise and iteratively reconstructs clean data:

$$p_\theta(x_{t-1} \mid x_t) = \mathcal{N}\left(x_{t-1}; \mu_\theta(x_t, t), \Sigma_\theta(x_t, t)\right).$$

The mean term $\mu_\theta(x_t, t)$ is computed from this noise prediction via a closed-form reparameterization: $\mu_\theta(x_t, t) = \frac{1}{\sqrt{\alpha_t}}\left(x_t - \frac{1-\alpha_t}{\sqrt{1-\bar{\alpha}_t}}\epsilon_\theta(x_t, t)\right)$, where $\alpha_t = 1 - \beta_t$ and $\bar{\alpha}_t = \prod_{s=1}^t \alpha_s$. The covariance $\Sigma_\theta(x_t, t)$ is typically fixed by the variance schedule $\{\beta_t\}$, though some variants allow it to be partially learned for improved sample quality (Nichol & Dhariwal, 2021). Together, $\mu_\theta$ and $\Sigma_\theta$ define the Gaussian reverse step, while $\epsilon_\theta$ remains the core predicted quantity that drives the denoising trajectory.

For adversarial purification, the intuition is to inject the adversarial input $x^{adv}$ into the forward process at a chosen noise level $t$, so that adversarial perturbations are drowned out by Gaussian noise (Nie et al., 2022). Then, the reverse process denoises $x_t$ step by step, ideally converging to a purified sample $x^{pur}$ close to the clean distribution. Formally, the purification mapping can be written as:

$$x^{pur} \sim P(x^{adv}) = p_\theta(x^{pur} \mid x_t^{adv}, t),$$
$$\text{with } x_t^{adv} \sim q(x_t \mid x^{adv}).$$

Here $x_t^{adv}$ denotes the adversarial input injected into the forward noising process at step $t$, and $x^{pur}$ is the purified output after reverse diffusion. This framework has achieved strong empirical robustness across various benchmarks (Nie et al., 2022; Wang et al., 2022; Chen et al., 2024b).

However, existing diffusion-based purification suffers from two main drawbacks: (i) the denoising process is essentially *unimodal*, since it is conditioned only on Gaussian noise schedules without leveraging explicit semantic cues (text prompts), which limits its ability to preserve class-consistent information; and (ii) the multi-step reverse process is computationally expensive, making such defenses inefficient for real-time deployment. These limitations motivate our proposed Multimodal Diffusion for Adversarial Purification with explicit prompt guidance and improved efficiency.

### 3.2. Stable Diffusion with Prompt Learning

While diffusion-based purification can remove adversarial perturbations, prior defenses typically rely on *small, unimodal* diffusion models to approximate the data distribution (Nie et al., 2022). Limited representational capacity often leads to suboptimal likelihood estimates and unstable denoising trajectories, where semantic information may not be faithfully preserved.

To address this limitation, we adopt Stable Diffusion, a large-scale latent diffusion model (LDM), as our backbone (Rombach et al., 2022). Pretrained on massive image–text corpora, Stable Diffusion provides substantially stronger modeling power and a richer, more informative likelihood landscape than small diffusion models. Moreover, operating in a compact latent space enables high-resolution synthesis with improved efficiency compared to pixel-space diffusion (Rombach et al., 2022; Dhariwal & Nichol, 2021). This stronger backbone lets our purifier start denoising from a more faithful approximation of the clean data manifold, reducing reliance on long reverse diffusion chains and mitigating semantic drift.

Formally, Stable Diffusion operates in a latent space defined by a variational autoencoder (VAE). Given an input image $x$, the encoder maps it into a compact latent representation $z = \mathcal{E}_{\text{VAE}}(x)$, where $\mathcal{E}_{\text{VAE}}$ denote the encoder. Given a class label or textual description $y$, we obtain a prompt embedding through a text encoder $e_p = \mathcal{E}_{\text{text}}(p)$, where $p$ is the input text, such as 'a photo of a cat'. In our approach, these embeddings serve as semantic conditions that guide the purification process. The denoising network $\epsilon_\theta(x_t, t, e_p)$ is implemented as a U-Net with cross-attention, which predicts the noise at each timestep. The denoising network $\epsilon_\theta(x_t, t, e_p)$ is trained with the standard denoising diffusion probabilistic model (DDPM) objective (Ho et al., 2020), which treats noise prediction as score matching:

$$\mathcal{L}_{\text{DDPM}}(\theta) = \mathbb{E}_{x_0, \epsilon, t}\left[\|\epsilon - \epsilon_\theta(x_t, t, e_p)\|_2^2\right].$$

where $x_t = \sqrt{\bar{\alpha}_t}x_0 + \sqrt{1 - \bar{\alpha}_t}\epsilon$, with $\epsilon \sim \mathcal{N}(0, I)$, $\alpha_t = 1 - \beta_t$, and $\bar{\alpha}_t = \prod_{s=1}^t \alpha_s$ denote the variance schedule. This loss enforces the network to accurately predict the added Gaussian noise at each timestep, which is equivalent to maximizing a variational lower bound on the conditional data likelihood. In our case, the conditioning $e_p$ provides semantic priors that explicitly align the denoising trajectory with the true class, thereby enhancing the stability and fidelity of purification.

**Prompt Learning Objective.** A central challenge for purification-based defenses lies in the accuracy of likelihood estimation during denoising. Although Stable Diffusion provides a strong backbone, its conditioning typically depends on fixed or manually designed text prompts, which may be generic and fail to provide task-specific guidance. Such limitations are particularly critical for adversarial purification, where the model must recover the clean data distribution from inputs corrupted by imperceptible but adversarial perturbations. To overcome this issue, we propose a *prompt learning* module that explicitly optimizes prompt embeddings from clean data, allowing the model to acquire semantic priors that are robust to adversarial noise.

In general, a prompt $p$ can be represented as a concatenation of $M$ learnable context tokens $p_{\text{context}} = [v_1, v_2, \ldots, v_M]$, where each $v_m \in \mathbb{R}^d$ has the same dimensionality as the text encoder's word embeddings (e.g., $d = 768$ for CLIP). In prior works, such context tokens are often combined with a class-specific token (e.g., the word "cat"), yielding a class-dependent prompt $p = [p_{\text{context}}, p_{\text{class}}]$ that provides label-conditioned guidance. By contrast, our objective is to design a *class-agnostic prompt* that captures global semantic priors without relying on class labels. This choice is crucial for adversarial purification, since the ground-truth label of an adversarial input is typically unknown at inference time. We therefore optimize a shared prompt vector $p$ directly from clean data, such that it enhances the unconditional likelihood estimation of the diffusion model.

Our prompt learning module is optimized by reusing the standard DDPM noise-prediction loss, with the key difference that only the prompt parameters $p$ are updated while the diffusion backbone $\theta$ remains frozen:

$$\mathcal{L}_{\text{prompt}}(p) = \mathbb{E}_{x_0, t, \epsilon}\Big[\; \big\| \epsilon - \epsilon_\theta\left(x_t, \; t, \; p\right) \big\|_2^2 \;\Big].$$

Optimizing this loss is equivalent to maximizing a variational lower bound on the likelihood $p_\theta(x_0 \mid p)$. Thus, the learned prompt $p^*$ serves as a universal semantic prior that stabilizes the denoising trajectory and improves the fidelity of adversarial purification without requiring class labels or adversarial training.

Similar to prompt learning in CLIP (Zhou et al., 2022), we optimize the learnable context tokens using a gradient-based method, such as Adam (Kingma, 2014). In each training iteration, we sample a mini-batch of clean data points $x^{(b)}$, apply the forward diffusion process to obtain noisy latents $x_t^{(b)}$ with Gaussian noise $\epsilon$, and evaluate the prompt loss $\mathcal{L}_{\text{prompt}}$ with the current tokens $p = [v_1, \ldots, v_M]$. The gradients are then backpropagated through the denoising network $\epsilon_\theta(x_t^{(b)}, t, p)$ to update the prompt parameters. This process is repeated until convergence, yielding a shared prompt vector $p^*$ that minimizes the denoising objective. The detailed optimization procedure is summarized in Algorithm 1. Compared to full model fine-tuning, optimizing only a small set of prompt parameters significantly reduces trainable variables, which mitigates overfitting and keeps computational cost manageable, while still providing strong semantic guidance for purification.

**Theoretical Guarantee.** We show that optimizing the class-agnostic prompt with the DDPM objective monotonically increases a variational lower bound (ELBO) of the unconditional data likelihood under a fixed diffusion model.

**Theorem 3.1** (Prompt learning improves the likelihood lower bound)**.** *Let $x_0 \sim p_{data}$ denote clean latents, and let $x_t = \sqrt{\bar{\alpha}_t}x_0 + \sqrt{1 - \bar{\alpha}_t}\,\epsilon$ with $\epsilon \sim \mathcal{N}(0, I)$. Fix the*

---

**Algorithm 1** Prompt Learning on Stable Diffusion

---

**input** Frozen diffusion backbone $\theta$ (VAE, U-Net $\epsilon_\theta$), clean images $\{x^{(b)}\}$, steps $T$, optimizer (Adam), prompt length $M$, iterations $N$
**output** Learned class-agnostic prompt $p^\star = [v_1, \ldots, v_M]$
 1: Initialize learnable tokens $p = [v_1, \ldots, v_M]$ (random or text-init)
 2: **for** $n = 1, \ldots, N$ **do**
 3:     Sample a mini-batch $\{x^{(b)}\}$, time $t \sim$ Unif$(\{1, \ldots, T\})$, and noise $\epsilon \sim \mathcal{N}(0, I)$
 4:     $x_t \leftarrow \sqrt{\bar{\alpha}_t}\, x^{(b)} + \sqrt{1 - \bar{\alpha}_t}\, \epsilon$
 5:     $\mathcal{L}_{\text{prompt}} \leftarrow \|\epsilon - \epsilon_\theta(x_t, p)\|_2^2$ {average over batch}
 6:     $p \leftarrow \text{Adam}\,(p, \nabla_p \mathcal{L}_{\text{prompt}})$ {keep $\theta$ frozen}
 7: **end for**
 8: **Return:** $p^\star \leftarrow p$

---

*diffusion backbone parameters $\theta$, and optimize only the prompt $p$ using the DDPM objective. Then the optimal prompt $p^\star = \arg\min_p \mathcal{L}_{prompt}(p)$ maximizes the evidence lower bound (ELBO) on the data likelihood $p_\theta(x_0 \mid p)$,*

$$\log \underline{p}_\theta(x_0 \mid p^\star) \; \geq \; \log \underline{p}_\theta(x_0 \mid p), \quad \forall p,$$

*where $\log \underline{p}_\theta$ denotes the variational lower bound.*

**Theorem 3.2** (Prompt learning provides a valid descent direction for the likelihood lower bound)**.** *Let $x_0 \sim p_{data}$ denote clean latents, and let $x_t = \sqrt{\bar{\alpha}_t} x_0 + \sqrt{1 - \bar{\alpha}_t}\,\epsilon$ with $\epsilon \sim \mathcal{N}(0, I)$. Fix the diffusion backbone parameters $\theta$, and optimize only the prompt $p$ using the DDPM objective $\mathcal{L}_{prompt}(p) = \frac{1}{T}\sum_{t=1}^{T}\mathbb{E}_{x_0, \epsilon}\big[\|\epsilon - \epsilon_\theta(x_t, t, p)\|_2^2\big]$. Let the variational lower bound objective be $\mathcal{L}_{VLB}(p) = \sum_{t=1}^{T} w_t \mathbb{E}_{x_0, \epsilon}\big[\|\epsilon - \epsilon_\theta(x_t, t, p)\|_2^2\big]$ with weights $w_t > 0$. If at a given $p$ the per-timestep gradients satisfy*

$$\cos\big(\nabla_p \mathcal{L}_{prompt}(p), \; \nabla_p \mathcal{L}_{VLB}(p)\big) > 0, \qquad (1)$$

*then for sufficiently small step size $\eta > 0$, updating $p$ along $-\nabla_p \mathcal{L}_{prompt}(p)$ strictly decreases $\mathcal{L}_{VLB}$:*

$$\mathcal{L}_{VLB}\big(p - \eta\,\nabla_p \mathcal{L}_{prompt}(p)\big) < \mathcal{L}_{VLB}(p). \qquad (2)$$

*That is, prompt learning with the uniform-weight DDPM objective monotonically improves the variational lower bound on $\log p_\theta(x_0 \mid p)$.*

**Corollary 3.3** (Score matching view)**.** *The objective $\mathcal{L}_{prompt}$ is equivalent to minimizing a weighted Fisher divergence between the conditional score $\nabla \log p_\theta(x_t \mid p)$ and the forward diffusion score $\nabla \log q(x_t \mid z_0)$. Hence optimizing $p$ directly improves the data likelihood $\log p_\theta(x_0 \mid p^\star) \; \geq \; \log p_\theta(x_0 \mid p), \forall p.$*

### 3.3. Prompt-Guided Likelihood Maximization for Purification

Given an adversarial input $x^{adv}$, we purify it by maximizing the model likelihood under the learned class-agnostic

prompt $p^\star$ while using the pretrained diffusion backbone $\theta$. We obtain a noisy image by the forward diffusion

$$x_{t^\star}^{adv} = \sqrt{\bar{\alpha}_{t^\star}}\, x_0^{adv} + \sqrt{1 - \bar{\alpha}_{t^\star}}\, \epsilon, \qquad \epsilon \sim \mathcal{N}(0, I).$$

Our goal is to recover an $x_0$ that maximizes the posterior (or the conditional likelihood surrogate)

$$x_0^{\mathrm{pur}} \in \arg\max_{x_0}\ \log p_\theta(x_{t^\star}^{adv} \mid x_0, p^\star) + \log p(x_0), \quad (3)$$

where $p(x_0)$ is the prior. Maximizing (3) is intractable directly, so we instead minimize the purification "simple loss" with respect to the image variable $x_0$, while conditioning on the learned prompt $p^\star$:

$$x_0^{\mathrm{pur}} \in \arg\min_{x_0}\ \underbrace{\mathbb{E}_{t,\epsilon} \left\| \epsilon - \epsilon_\theta(x_t, t, p^\star) \right\|_2^2}_{=: \mathcal{L}_{\mathrm{DDPM}}(x_0; p^\star)} + \lambda\, \mathcal{R}(x_0, x^{adv}),$$
$$(4)$$

where $t$ is sampled uniformly from $\{1, \ldots, T\}$, $\epsilon \sim \mathcal{N}(0, I)$, and $\mathcal{R}$ is an optional proximity or naturalness regularizer (e.g., $\mathcal{R}(x_0, x^{adv}) = \|x_0 - x^{adv}\|_2^2$ or a TV prior). By Theorem 3.1, minimizing $\mathcal{L}_{\mathrm{DDPM}}$ w.r.t. $p$ tightens the ELBO; when optimizing w.r.t. $x_0$, Eq. (4) serves as a surrogate that increases the conditional likelihood under $p^\star$.

**Gradient and Update.** Let $x_t(x_0, \epsilon) = \sqrt{\bar{\alpha}_t}\, x_0 + \sqrt{1 - \bar{\alpha}_t}\, \epsilon$. The gradient of Eq. (4) is

$$\nabla_{x_0} \mathcal{L}_{\mathrm{Pur}} = \nabla_{x_0} \mathbb{E}_{t,\epsilon} \left[ \sqrt{\bar{\alpha}_t}\, \nabla_{x_t} \left\| \epsilon - \epsilon_\theta(x_t, t, p^\star) \right\|_2^2 \right]$$
$$+ \lambda\, \nabla_{x_0} \mathcal{R}(x_0, x^{adv}),$$

where we define $\mathcal{L}_{\mathrm{Pur}} = \mathcal{L}_{\mathrm{DDPM}} + \lambda\, \mathcal{R}(x_0, x^{adv})$ and we perform a few iterations of gradient descent with box constraints:

$$x_0^{(k+1)} = \Pi_{[0,1]} \left( x_0^{(k)} - \eta\, \nabla_{x_0} \mathcal{L}_{\mathrm{Pur}}(x_0^{(k)}; p^\star) \right), x_0^{(0)} = x^{adv},$$

where $\Pi_{[0,1]}$ clips pixels to the valid range and $\eta$ is the step size. In practice, we estimate the expectations with a single $(t, \epsilon)$ per iteration and use 5–10 steps; the prompt guidance $p^\star$ stabilizes the descent by injecting high-level semantics, yielding fast and faithful purification in pixel space. Besides, we adopt the proximity regularizer $\mathcal{R}(x_0, x^{adv}) = \|x_0 - x^{adv}\|_2^2$ with a fixed weight $\lambda = 0.9$, which encourages purified outputs to remain close to the original adversarial inputs while removing perturbations. The overall purification process is summarized in Algorithm 2.

**Theoretical Guarantee**    We analyze the pixel–space purification objective

$$\mathcal{L}_{\mathrm{pur}} = \mathcal{L}_{\mathrm{DDPM}}(x_0; p^\star) + \lambda \mathcal{R}(x_0, x^{adv}). \qquad (5)$$

**Algorithm 2** Purification via Regularized DDPM-Loss Minimization in Pixel Space

---

**input**  Adversarial image $x^{adv}$, learned prompt $p^\star$, frozen $\theta$, steps $T_1\ T_2$, Purification steps $N$, step size $\eta$, optional regularizer weight $\lambda$.

**output**  Purified image $x^{\mathrm{pur}}$. $x_0^{(0)} \leftarrow x^{adv}$

1: **for** $n = 0, \ldots, N - 1$ **do**
2:      Sample $t \sim \mathrm{Unif}(\{T_1, \ldots, T_2\})$ and $\epsilon \sim \mathcal{N}(0, I)$.
3:      $x_t \leftarrow \sqrt{\bar{\alpha}_t}\, x_0^{(n)} + \sqrt{1 - \bar{\alpha}_t}\, \epsilon$
4:      $\mathcal{L}_{\mathrm{pur}} \leftarrow \left\| \epsilon - \epsilon_\theta(x_t, t, p^\star) \right\|_2^2 + \lambda \mathcal{R}\left( x_0^{(n)}, x^{adv} \right)$
5:      $g \leftarrow \nabla_{x_0} \mathcal{L}_{\mathrm{pur}}$
6:      $x_0^{(n+1)} \leftarrow \Pi_{[0,1]}\left( x_0^{(n)} - \eta\, g \right)$
7: **end for**
8: $x^{\mathrm{pur}} \leftarrow x_0^{(N)}$
9: **Return:** $x^{\mathrm{pur}}$.

---

Let $x_t(x_0, \epsilon) = \sqrt{\bar{\alpha}_t} x_0 + \sqrt{1 - \bar{\alpha}_t}\epsilon$ and denote $\ell(x_0; t, \epsilon) = \left\| \epsilon - \epsilon_\theta(z_t, t, p^\star) \right\|_2^2$. We update $x_0$ by projected SGD with one sampled $(t, \epsilon)$ per step:

$$x_0^{(k+1)} = \Pi_{[0,1]} \left( x_0^{(k)} - \eta\, g(x_0^{(k)}; t_k, \epsilon_k) \right),$$
$$g(x_0; t, \epsilon) = \nabla_{x_0} \left( \ell(x_0; t, \epsilon) + \lambda \mathcal{R}(x_0, x^{adv}) \right).$$

**Assumptions.**  (A1) *Smoothness:* $\mathcal{L}_{\mathrm{Pur}}(x_0; p^\star)$ is $L$–smooth on $[0, 1]^d$. (A2) *Bounded variance:* $\mathbb{E}\left[ \| g(x_0; t, \epsilon) - \nabla \mathcal{L}_{\mathrm{Pur}}(x_0; p^\star) \|_2^2 \right] \leq \sigma^2$. (A3) *Regularizer:* $\mathcal{R}$ is convex and $L_R$–smooth (e.g., $\|x_0 - x^{adv}\|_2^2$ or TV with smooth surrogate).

**Lemma 3.4** (Unbiased one–sample gradient with bounded variance). *With the reparameterization $z_t(x_0, \epsilon)$, the stochastic gradient is unbiased:*

$$\mathbb{E}_{t,\epsilon}\left[ g(x_0; t, \epsilon) \right] = \nabla_{x_0} \mathcal{L}_{Pur}(x_0; p^\star),$$

*and satisfies Assumption (A2).*

*Proof Sketch.* Differentiate under the expectation using reparameterization; the Jacobian $\partial x_t / \partial x_0 = \sqrt{\bar{\alpha}_t} I$ is deterministic. Linearity of expectation and the uniform sampling of $t$ give the unbiasedness. Bounded variance follows from (A1) and standard Lipschitz/activation bounds on $\epsilon_\theta$. □

**Theorem 3.5** (Expected descent of the purification loss). *Under (A1)–(A3), let $\eta \leq 1/(2L)$. Then the projected SGD iterate satisfies*

$$\mathbb{E}\left[ \mathcal{L}_{Pur}(x_0^{(k+1)}; p^\star) \right] \leq \mathbb{E}\left[ \mathcal{L}_{Pur}(x_0^{(k)}; p^\star) \right]$$
$$- \frac{\eta}{2} \mathbb{E}\left[ \| \nabla \mathcal{L}_{Pur}(x_0^{(k)}; p^\star) \|_2^2 \right] + \frac{\eta^2 L}{2}\, \sigma^2.$$

*Consequently, after K steps,*

$$\frac{1}{K}\sum_{k=0}^{K-1}\mathbb{E}\left[\|\nabla\mathcal{L}_{DDPM}(x_0^{(k)};p^\star)\|_2^2\right] \leq$$

$$\frac{2(\mathcal{L}_{DDPM}(x_0^{(0)};p^\star) - \mathcal{L}_{inf})}{\eta K} + \eta L\sigma^2,$$

*where $\mathcal{L}_{inf}$ is the infimum.*

*Proof Sketch.* Apply the standard smoothness (descent) lemma to the projected step and take expectations. Use Lemma 3.4 to replace $\mathbb{E}[g]$ with the true gradient and bound the variance term by $\sigma^2$. Summing the per–step inequality yields the average–gradient bound. $\qquad\square$

We further verify the practical validity of the assumptions used in Theorem 3.5 through empirical measurements of the local Lipschitz constant and gradient variance; see Appendix A.3 for details.

**Why Single $(t, \epsilon)$ and Few Steps Suffice.** The one–sample estimator is unbiased but noisy; this *stochasticity* serves as exploration that helps escape local pixel–level artifacts, while Theorem 3.5 guarantees expected descent provided $\eta$ is small. Moreover, the bound shows $O(1/K)$ decay of the average gradient norm up to a variance floor $\eta L\sigma^2$, so a *small fixed* number of steps ($N = 5$–$20$) already yields a measurable reduction of the loss/ELBO gap—matching our practice.

## 4. Experiments

### 4.1. Experimental Design

This section presents an extensive empirical study to validate the effectiveness of the proposed Multimodal Diffusion for Adversarial Purification (MultiDAP). We describe experimental setups (datasets and implementation), report quantitative and qualitative results under adversarial attacks, and provide ablations dissecting the contributions of different design choices.

**Datasets and Model Architectures.** We evaluate our method on three standard benchmarks: CIFAR-10, CIFAR-100, and ImageNet-1K. All input images are resized to 256 × 256 to match the input resolution of Stable Diffusion. For CIFAR-10 and CIFAR-100, we utilize their full clean training sets (50,000 images) to optimize the class-agnostic prompt embeddings. For the large-scale ImageNet-1K, to demonstrate data efficiency and scalability, we optimize the prompt using only a randomly sampled subset of 60,000 training images. We evaluate purification performance on a held-out subset of 512 adversarial test images sampled from the standard test split for each dataset. We use miniSD-diffusers (Lambda Labs, 2022) as a frozen stable diffusion generative backbone for purification. By default we adopt a class-agnostic prompt, parameterized as a set of $M$ learnable context tokens that are shared across all images and classes. For ablation studies, we also evaluate hand-crafted class-agnostic prompts (e.g., "a photo of"), which provide weaker guidance compared to our learned prompt but highlight the effectiveness of explicitly optimizing context tokens. The classifiers are WideResNet70-16, WideResNet-28-10, and ResNet-50 for CIFAR-10, CIFAR-100, and ImageNet, respectively. All classifiers are pretrained on clean datasets.

**Implementation Details.** We initialize miniSD-diffuser and freeze all network parameters except for the learnable prompt embeddings. During prompt learning, we optimize learnable tokens of dimension $d = 768$ using the Adam optimizer where the learning rate is $2 \times 10^{-4}$. For CIFAR datasets, we set the prompt length $M = 16$ and use a batch size of 64, training for 10 epochs. For ImageNet, we increase the prompt capacity to $M = 64$ and use a batch size of 8, training for 1 epoch. We incorporate an expectation of the noise $\epsilon$ and timestep $t$ while training, ensuring consistency with the diffusion objective. Note that adversarial attacks are imposed to input images. The learned prompt $p^\star$ is not attacked or optimized during the attacking and purification process. During purification, we inject adversarial examples into the forward diffusion process with a single randomly sampled timestep $t \sim \text{Unif}([T_1, T_2])$ and Gaussian noise $\epsilon \sim \mathcal{N}(0, I)$, where we set $T_1 = 400$ and $T_2 = 600$. We then run gradient descent on $x_0$ using the Regularized DDPM loss (Eq. (5)), which provides an efficient surrogate for likelihood maximization. The step size $\eta$ is set to 0.2 for CIFAR datasets and 0.5 for ImageNet, and we clip pixel values into $[0, 1]$ after each update. Unless otherwise specified, we adopt the class-agnostic prompt $p^\star$ learned in Sec. 3.2 as the conditioning signal.

**Adversarial Attacks.** We evaluate robustness against two widely used adversarial attacks for diffusion-based adversarial purification. The first is PGD-20 (Madry et al., 2018), a multi-step iterative $\ell_\infty$-bounded attack with random restarts. The second is AutoAttack (Croce & Hein, 2020), a parameter-free ensemble of diverse attacks. Because of the stochasticity in MultiDAP, we use rand version of AutoAttacks. Following (Chen et al., 2024b)'s settings, we have n_iter = 100 for AutoAttack. Unless otherwise specified, the maximum perturbation budget is set to $\epsilon = 8/255$ for $\ell_\infty$ threat models and $\epsilon = 0.5$ for $\ell_2$ threat models.

**Baselines.** We compare our method with two representative purification-based defenses. The first is DiffPure (Nie et al., 2022), which performs iterative reverse diffusion to

*Table 1.* Clean and robust accuracy (%) on CIFAR-10. Robust results under AutoAttack are reported for $\ell_\infty$ ($\epsilon = 8/255$) and $\ell_2$ ($\epsilon = 0.5$). The last column reports $\ell_\infty$ PGD-20 with 10 random restarts (step size $\alpha = \epsilon/4$). For DiffPure, $t_1 = 0.125$ and $t_2 = 0.1$ denote the time scales used. Our method (MultiDAP) uses a class-agnostic prompt and only 5 purification steps. We also report an ablation using the fixed template prompt "a photo of a", which is widely used in CLIP-based zero-shot classification. **Bold** denotes the best, underline the second best, and shading the third best.

| Method | Architecture | Clean Acc | AA ($\ell_\infty$) | AA ($\ell_2$) | PGD-20 |
|---|---|---|---|---|---|
| AT-DDPM-$\ell_\infty$ | WRN28-10 | 88.87 | 63.28 | 64.65 | 55.31 |
| AT-DDPM-$\ell_2$ | WRN28-10 | 93.16 | 49.41 | 81.05 | 51.47 |
| AT-EDM-$\ell_\infty$ | WRN28-10 | 93.36 | 70.90 | 69.73 | **72.96** |
| AT-EDM-$\ell_2$ | WRN28-10 | 95.90 | 53.32 | **84.77** | 55.34 |
| DiffPure ($t_1$) | UNet+WRN70-16 | 87.50 | 40.62 | 75.59 | 47.89 |
| DiffPure ($t_2$) | UNet+WRN70-16 | 90.97 | 44.53 | 72.65 | 51.89 |
| LM | UNet+WRN70-16 | 87.89 | 71.68 | 75.00 | 65.22 |
| CLIPure-Diff | CLIP-ViT-L/14 | 93.75 | 55.74 | 80.02 | 58.24 |
| CLIPure-Cos | CLIP-ViT-L/14 | 84.38 | 64.21 | 65.94 | 66.41 |
| OSCP (CAP-only) | SD1.5+WRN70-16 | **96.60** | 66.45 | 82.02 | 58.60 |
| MultiDAP ("a photo of a ·") | UNet+WRN70-16 | 93.80 | 70.29 | 74.53 | 65.00 |
| MultiDAP (prompt learning) | UNet+WRN70-16 | 94.12 | **72.38** | 76.05 | 68.21 |

*Table 2.* Clean and robust accuracy (%) on ImageNet-1K. Robust results under AutoAttack are reported for $\ell_\infty$ ($\epsilon = 4/255$) on 512 randomly selected test samples. Our method (MultiDAP) uses a class-agnostic prompt and 20 purification steps. We also report an ablation using the fixed template prompt "a photo of a". **Bold** denotes the best, underline the second best, and shading the third best.

| Method | Architecture | Clean Acc | AA |
|---|---|---|---|
| AT (Engstrom et al., 2019) | ResNet-50 | 62.56 | 31.06 |
| Fast AT (Wong et al., 2020) | ResNet-50 | 55.62 | 26.95 |
| Strong PGD AT (Salman et al., 2020) | ResNet-50 | 64.02 | 37.89 |
| AT (Bai et al., 2021) | ResNet-50 | 67.38 | 35.51 |
| DiffPure ($t = 0.15$) | UNet+ResNet-50 | **67.79** | 40.93 |
| LM | UNet+ResNet-50 | 66.51 | 38.27 |
| MultiDAP ("a photo of a ·") | UNet+ResNet-50 | 66.25 | 38.65 |
| MultiDAP (prompt learning) | UNet+ResNet-50 | 67.14 | **41.26** |

remove adversarial perturbations. The second is Likelihood Maximization (LM) (Chen et al., 2024b) which formulates purification as direct maximization of the unimodal diffusion model likelihood. The third is CLIPure (Zhang et al., 2025), a CLIP-based zero-shot purification method with two variants: CLIPure-Diff, which estimates image likelihood via the generative latent process, and CLIPure-Cos, which measures likelihood using the cosine similarity between image embeddings and a blank template. These methods provide strong and conceptually related baselines for evaluating the effectiveness of our proposed MultiDAP. Other baseline methods are adversarial training methods which rely on unimodal diffusion models to generate adversarial examples or argument.

### 4.2. Robustness against Adversarial Attacks

Table 1 summarizes the robustness results of our method on CIFAR-10 compared with DiffPure and Likelihood Maximization under PGD-20 and AutoAttack. Our proposed MultiDAP achieves substantially better performance than Likelihood Maximization across both $\ell_\infty$ and $\ell_2$ threat models, highlighting the benefit of leveraging a learned prompt prior. Compared with DiffPure, MultiDAP attains slightly higher robust accuracy, but requires only 5 purification steps, whereas DiffPure typically involves dozens of Langevin iterations. This efficiency gap makes MultiDAP significantly more practical in scenarios where the inference speed is critical, while still preserving competitive robustness. We leave the results for CIFAR-100 in Appendix A.4.

Table 2 demonstrates the scalability of our method on the large-scale ImageNet-1K dataset. Consistent with the results on CIFAR benchmarks, MultiDAP achieves state-of-the-art robustness while maintaining high efficiency with only 20 purification steps. Our prompt-learning variant attains 41.26% robust accuracy under AutoAttack, significantly outperforming strong adversarial training baselines such as Strong-AT (Salman et al., 2020) (37.89%) and matching the heavy-computation DiffPure (Nie et al., 2022) (40.93%).

*Table 3.* Computational cost comparison under the same perturbation budget using AutoAttack. FLOPs and inference time are measured per image on an NVIDIA A100. "FLOPs (G)" represents the total number of floating-point operations required for full pipeline. "Time (ms)" reports the end-to-end inference latency measured at batch size 1. "Prompt (s)" indicates prompt-learning cost. The FLOPs difference of DiffPure between CIFAR-10 and CIFAR-100 arises from its use of different backbone classifiers (WideResNet-70-16 for CIFAR-10 and WideResNet-28-10 for CIFAR-100).

| | CIFAR-10 | | | CIFAR-100 | | |
| --- | --- | --- | --- | --- | --- | --- |
| Method | FLOPs (G) | Time (ms) | Prompt (s) | FLOPs (G) | Time (ms) | Prompt (s) |
| DiffPure | 212.4 | 1567.40 | – | 34.87 | 1520.38 | – |
| CLIPure | 51.9 | 410.50 | – | 12.16 | 402.74 | – |
| LM | 43.9 | 385.26 | – | 7.25 | 376.31 | – |
| MultiDAP | **38.8** | **371.48** | 5800 | **5.25** | **361.95** | 43200 |

*Table 4.* Ablation studies on CIFAR-10 under AutoAttack ($\ell_\infty$-norm $\epsilon = 8/255$, n_iter = 100). Left: effect of purification steps $N$ with fixed $[T_1, T_2] = (400, 600)$ and $\eta = 0.2$. Right: effect of timestep range $[T_1, T_2]$ with fixed $N = 5$ and $\eta = 0.2$.

| $N$ | $\eta$ | Clean Acc | Robust Acc |
| --- | --- | --- | --- |
| 1 | 0.2 | **95.01** | 62.31 |
| 2 | 0.2 | 94.57 | 65.24 |
| 5 | 0.2 | 94.12 | **72.38** |
| 10 | 0.2 | 91.31 | 71.88 |
| 20 | 0.2 | 92.88 | 65.12 |
| 30 | 0.2 | 88.75 | 58.44 |
| 50 | 0.2 | 85.94 | 53.75 |

| $[T_1, T_2]$ | $N$ | $\eta$ | Clean Acc | Robust Acc |
| --- | --- | --- | --- | --- |
| $[100, 200]$ | 5 | 0.2 | **96.12** | 58.44 |
| $[200, 400]$ | 5 | 0.2 | 95.31 | 65.32 |
| $[400, 600]$ | 5 | 0.2 | 94.12 | **72.38** |
| $[600, 800]$ | 5 | 0.2 | 92.25 | 66.41 |
| $[800, 999]$ | 5 | 0.2 | 90.14 | 60.27 |

This confirms that leveraging rich cross-modal semantic information is a key factor for effective purification on high-resolution, diverse natural images.

While prompt engineering relies on manually crafted textual templates (e.g., "a photo of a ·"), our approach learns task-adaptive prompt embeddings that encode richer semantic priors. This learned representation provides stronger and more flexible conditioning, enabling the diffusion model to better suppress adversarial noise and reconstruct class-consistent images even under severe perturbations. Compared to hand-designed prompts, prompt learning thus offers a systematic and scalable way to inject semantic guidance into the purification process.

Prompt learning on CIFAR in MultiDAP is lightweight: only $16 \times 768$ learnable tokens ($< 0.02\%$ of Stable Diffusion's parameters) introduced and converges within 2 hours on 1 A100 GPU for CIFAR-10. As in Table 3, MultiDAP is the most computationally efficient method across both CIFAR-10 and CIFAR-100. It achieves the lowest FLOPs and fastest inference latency, i.e., 38.8 G FLOPs and 371.48 ms on CIFAR-10; 5.25 G FLOPs and 361.95 ms on CIFAR-100, while maintaining competitive robustness. Although MultiDAP includes a one-time offline prompt-learning stage, its test-time cost remains substantially lower than DiffPure and CLIPure, highlighting its practicality for real-world deployment.

From a theoretical perspective, the robustness of MultiDAP

can be explained by the stochastic nature of purification combined with the learned semantic prior. Instead of explicitly computing the expectation over all timesteps and noises, which would be computationally prohibitive, we approximate it with a single random $(t, \epsilon)$ per iteration. This stochastic approximation, when combined with semantic guidance from prompts, provides both efficiency and regularization, preventing overfitting to specific perturbations. In practice, this explains why only 5–10 purification steps are sufficient to achieve competitive robustness, which is consistent with our ablation results showing that excessive steps may even degrade performance (see Table 4).

### 4.3. Ablation Studies

**Number of Purification Steps.** We vary the number of purification steps $N$ with timestep range $[400, 600]$ and step size $\eta = 0.2$. Table 4 (left) shows robustness improves from $N = 1$ to 5, peaking at $N = 5$. Using $N = 10$ is comparable, while larger $N$ yields diminishing gains and can reduce both clean and robust accuracy, likely due to overfitting to injected noise. Overall, $N \in [5, 10]$ offers a strong robustness–efficiency trade-off, suggesting that a short optimization trajectory is sufficient to remove adversarial perturbations.

**Choice of Timestep Range.** We vary the timestep range $[T_1, T_2]$ with $N = 5$ and $\eta = 0.2$. Table 4 (right) shows small ranges (e.g., $[100, 200]$) inject insufficient noise and

yield weaker robustness, while large ranges (e.g., $[800, 999]$) oversmooth reconstructions and reduce clean accuracy. Mid-range choices (e.g., $[400, 600]$) perform best, balancing perturbation removal and semantic fidelity.

Appendices A.1 and A.2 present more ablation studies on prompt length $M$ and regularization strength $\lambda$, respectively.

## 5. Conclusions and Discussions

We proposed Multimodal Diffusion for Adversarial Purification (MultiDAP), a novel adversarial defense that leverages text-to-image diffusion models guided by learnable prompts. By injecting multimodal semantic priors through the text encoder, MultiDAP guides purification toward label-consistent reconstructions without adversarial training or distillation of the diffusion model. Empirically, across CIFAR-10, CIFAR-100, and ImageNet-1K, MultiDAP matches or improves upon diffusion-based purification baselines while requiring only a small number of purification iterations (e.g., 5–20), yielding a practical robustness–efficiency trade-off. On the theory side, we provide guarantees for both stages and connect them to practice via analysis of optimization stability, supporting the effectiveness of short-step purification guided by learned prompts.

**Limitations and Future Work.** Despite promising results, MultiDAP currently depends on Stable Diffusion backbones, which are computationally heavier than standard feed-forward models. Future work may explore lightweight diffusion architectures or truncated sampling to further improve efficiency. Moreover, extending MultiDAP domain-shifted benchmarks would provide stronger evidence of generalization. Another exciting direction is to incorporate richer or multi-modal prompts (e.g., textual descriptions beyond class names) to enhance semantic guidance during purification.

## Acknowledgements

This research was supported by JSPS KAKENHI Grant Number JP23K28109 and JSPS Bilateral Program Number JPJSBP120257420. MB was partially supported by JSPS KAKENHI Grant Number JP25K21288. WH was supported by JSPS KAKENHI Grant Number JP24K20848 and JST BOOST Grant Number JPMJBY24G6. AW was partially supported by JSPS KAKENHI Grant Number JP25K21283. CL was partially supported by JSPS KAKENHI Grant Number JP24K03005.

## Impact Statement

This paper presents work whose goal is to advance the field of Machine Learning. There are many potential societal consequences of our work, none which we feel must be specifically highlighted here.

This research employs computational approaches exclusively with publicly accessible datasets, avoiding any human subject involvement or confidential data handling. We adhere to ICML's ethical guidelines without competing interests, emphasizing responsible deployment of our contributions while ensuring transparent reporting to support reproducible research practices.

Our experiments utilize publicly available datasets with detailed experimental configurations provided throughout the paper. To facilitate reproducibility, we commit to releasing the complete source code upon paper acceptance, enabling the research community to validate and build upon our findings.

In this work, large language models (LLM) were used to polish its writing, and correct the grammar and spelling. We strictly follow ICML's guidelines and ethics for LLM usage.

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

# A. More Experiments

The code of our experiments is available at https://github.com/mbai8854/MultiDAP.

## A.1. Ablation Study on Prompt Length

To further understand the effect of prompt capacity, we conduct an ablation study by varying the prompt length $M$ while fixing the regularization strength $\lambda = 0.9$ and step size $\eta = 0.2$. The results are shown in Table 6 (left). We observe that both clean accuracy and robustness generally improve as $M$ increases from 4 to 16, suggesting that a moderate number of context tokens is beneficial for encoding richer semantic priors that guide the purification dynamics. The best robustness (72.38% AutoAttack under $\ell_\infty$) is achieved at $M = 16$. Increasing the prompt length further to 32 does not provide additional gains and even slightly reduces robustness, likely because overly long prompts introduce redundancy that makes optimization more difficult. These findings indicate that a compact yet expressive prompt length is sufficient for effective semantic conditioning.

## A.2. Ablation Study on Regularization Strength

We further investigate the impact of the regularization strength $\lambda$ while fixing the prompt length $M = 16$ and $\eta = 0.2$. The results in Table 6 (right) show that robustness improves steadily as $\lambda$ increases from 0.0 to 1.5, with the best performance (72.38% AutoAttack under $\ell_\infty$) obtained at $\lambda = 0.9$. This trend suggests that regularizing the prompt embeddings is important for preventing overfitting to individual noise realizations, thereby stabilizing the purification process. However, very large regularization values (e.g., $\lambda = 1.5$) begin to degrade robustness, indicating that excessive constraint may limit the representational flexibility. Overall, these results demonstrate that an intermediate regularization level provides the best trade-off between stability and semantic expressiveness.

## A.3. Empirical Estimation of Lipschitz Constant and Gradient Variance

To verify the practical assumptions required by Theorem 3.5, we conduct an empirical evaluation of the local Lipschitz constant $L$ and gradient variance $\sigma$ of the purification objective $g(x) = \nabla_x L_{\text{pur}}(x)$, where $L_{\text{pur}}$ is the DDPM-based denoising loss used in our purification step. Both measurements strictly follow the forward process in our method, including VAE encoding, DDPM noise injection, and UNet denoising.

First, we estimate the local Lipschitz constant via a finite-difference approximation: $L(x) \approx \frac{\|g(x+\delta)-g(x)\|_2}{\|\delta\|_2}$, where $\delta$ is a small random perturbation ($\|\delta\|_2 \approx 10^{-3}$). We repeat this procedure for multiple random perturbations and multiple images. To estimate $\sigma$, we sample gradients under random diffusion timesteps $t \sim \mathcal{U}[t_{\text{st}}, t_{\text{ed}}]$ and random noise $\epsilon \sim \mathcal{N}(0, I)$: $\sigma^2 = \text{Var}_{t,\epsilon}[g(x)]$. All gradients are computed in pixel space to match the theoretical setting.

Across CIFAR–10 samples, we obtain the empirical statistics shown in Table 5. These values are small, stable, and tightly bounded, confirming that the assumptions in Theorem 3.5 hold in practice. In particular, the purification gradient exhibits Lipschitz-smooth behavior and extremely low stochastic variance. This explains why a small number of purification steps (five iterations) is numerically stable and effective in our method.

| Metric | Mean | Max / Std |
|---|---|---|
| Lipschitz constant $L$ | 0.1724 | 0.3000 |
| Gradient std. $\sigma$ | $1.05 \times 10^{-4}$ | — |

*Table 5.* Empirical estimates of the Lipschitz constant and gradient variance of the purification loss. Small and stable values indicate smooth gradient behavior and validate the assumptions in Theorem 3.5.

## A.4. CIFAR-100

Table 7 reports results on the more challenging CIFAR-100 dataset. Consistent with the CIFAR-10 findings, MultiDAP achieves strong robustness while using only 5 purification steps. Our prompt-learning variant obtains 38.15% AutoAttack robustness, outperforming all diffusion-based baselines, including DiffPure (Nie et al., 2022) and Diffusion+Contrastive (Bai et al., 2024). Notably, even the fixed template prompt ("a photo of a") already provides competitive performance (35.70% AA), demonstrating that semantic conditioning—whether learned or hand-designed—substantially benefits the

*Table 6.* Ablation studies on CIFAR-10 under AutoAttack ($\ell_\infty$-norm $\epsilon = 8/255$). Left: effect of prompt length $M$ with fixed regularization strength $\lambda = 0.9$ and $\eta = 0.2$. Right: effect of regularization strength $\lambda$ with fixed prompt length $M = 16$ and $\eta = 0.2$.

| Prompt length | Clean Acc | AA ($\ell_\infty$) |
|---|---|---|
| 4 | 90.87 | 71.19 |
| 8 | 92.31 | 72.26 |
| 12 | 91.56 | 72.25 |
| 16 | 94.12 | 72.38 |
| 32 | 93.75 | 72.04 |

| Regularization strength | Clean Acc | AA ($\ell_\infty$) |
|---|---|---|
| 0.0 | 90.91 | 70.17 |
| 0.3 | 92.56 | 71.25 |
| 0.6 | 93.72 | 72.34 |
| 0.9 | 94.12 | 72.38 |
| 1.2 | 94.38 | 71.01 |
| 1.5 | 95.10 | 69.55 |

*Table 7.* Clean and robust accuracy (%) on CIFAR-100. Robust results under AutoAttack are reported for $\ell_\infty$ ($\epsilon = 8/255$). Our method (MultiDAP) uses a class-agnostic prompt and 5 purification steps. We also report an ablation using the fixed template prompt "a photo of a". **Bold** denotes the best, underline the second best, and shading the third best.

| Method | Architecture | Clean Acc | AA |
|---|---|---|---|
| AT-CutMix (Rebuffi et al., 2021) | WRN28-10 | 62.97 | 29.80 |
| AT-DDPM (Rebuffi et al., 2021) | WRN28-10 | 59.18 | 30.81 |
| AT-DDPM + CutMix (Rebuffi et al., 2021) | WRN28-10 | 62.41 | 32.06 |
| AT-DDPM (Pang et al., 2022) | WRN28-10 | 62.08 | 31.40 |
| AT-DDPM (Wang et al., 2023) | WRN28-10 | 72.58 | **38.83** |
| DiffPure ($t = 0.1$) (Nie et al., 2022) | UNet+WRN28-10 | 62.50 | 8.60 |
| Diffusion + Contrastive ($t = 0.1$) (Bai et al., 2024) | UNet+WRN28-10 | 57.82 | 24.22 |
| LM (Chen et al., 2024b) | UNet+WRN28-10 | 66.45 | 33.83 |
| MultiDAP ("a photo of a ·") | UNet+WRN28-10 | **73.29** | 35.70 |
| MultiDAP (prompt learning) | UNet+WRN28-10 | 72.52 | 38.15 |

purification dynamics. These results indicate that MultiDAP generalizes effectively to datasets with larger label spaces and more fine-grained visual variability.

## B. Complete proofs

### B.1. Proof of Theorem 3.1

*Proof of Theorem 3.1.* For DDPM (or latent DDPM), the negative ELBO can be written (Ho et al., 2020; Nichol & Dhariwal, 2021) as

$$-\log p_\theta(z_0 \mid p) = C(\theta) + \sum_{t=1}^{T} \underbrace{\mathbb{E}\left[\mathrm{KL}\big(q(z_{t-1} \mid z_t, z_0) \,\|\, p_\theta(z_{t-1} \mid z_t, p)\big)\right]}_{=:\ \mathcal{R}_t(p)} + \underbrace{\mathbb{E}\left[\mathrm{KL}\big(q(z_T) \,\|\, p(z_T)\big)\right]}_{\text{independent of } p}, \quad (6)$$

where $C(\theta)$ is independent of $p$ and only $\mathcal{R}_t(p)$ depends on the prompt.

Using the standard mean parameterization,

$$\mu_\theta(x_t, t, p) = \frac{1}{\sqrt{\alpha_t}}\Big(x_t - \frac{1 - \alpha_t}{\sqrt{1 - \bar{\alpha}_t}}\,\epsilon_\theta(x_t, t, p)\Big), \quad \Sigma_\theta(x_t, t) = \sigma_t^2 I,$$

one can show that $\mathcal{R}_t(p)$ is equivalent to a weighted noise-prediction MSE:

$$\mathcal{R}_t(p) = w_t\,\mathbb{E}_{x_0, t, \epsilon}\left[\big\|\epsilon - \epsilon_\theta(x_t, t, p)\big\|_2^2\right] + \text{const}, \quad w_t = \frac{\beta_t^2}{2\sigma_t^2 \alpha_t (1 - \bar{\alpha}_t)} > 0.$$

Substituting into Eq. (6) yields

$$\mathcal{L}_{\mathrm{VLB}}(p) =: -\log p_\theta(x_0 \mid p) = C'(\theta) + \sum_{t=1}^{T} w_t\,\mathbb{E}_{x_0, t, \epsilon}\left[\big\|\epsilon - \epsilon_\theta(x_t, t, p)\big\|_2^2\right],$$

which differs from $\mathcal{L}_{\text{prompt}}(p)$ only by positive weights and a constant. Therefore,

$$\arg\min_p \mathcal{L}_{\text{prompt}}(p) = \arg\min_p \mathcal{L}_{\text{VLB}}(p).$$

Let $p^\star$ be the minimizer; then $\mathcal{L}_{\text{VLB}}(p^\star) \le \mathcal{L}_{\text{VLB}}(p)$ for all $p$, equivalently $\log \underline{p}_\theta(x_0 \mid p^\star) \ge \log \underline{p}_\theta(x_0 \mid p)$. □

## B.2. Proof of Lemma 3.4

*Proof of Lemma 3.4.* We first recall the purification objective:

$$\mathcal{L}_{\text{pur}}(x_0; p^\star) = \mathbb{E}_{t,\epsilon}\big[\ell(x_0; t, \epsilon)\big] + \lambda \mathcal{R}(x_0, x^{adv}),$$

where $t \sim \text{Unif}(\{1, \dots, T\})$ and $\epsilon \sim \mathcal{N}(0, I)$.

By linearity of expectation and interchangeability of expectation and gradient under standard regularity conditions:

$$\nabla_{x_0} \mathcal{L}_{\text{pur}}(x_0; p^\star) = \nabla_{x_0} \mathbb{E}_{t,\epsilon}\big[\ell(x_0; t, \epsilon)\big] + \lambda \nabla_{x_0} \mathcal{R}(x_0, x^{adv}).$$

On the other hand, the stochastic gradient $g(x_0; t, \epsilon)$ is defined as

$$g(x_0; t, \epsilon) = \nabla_{x_0} \ell(x_0; t, \epsilon) + \lambda \nabla_{x_0} \mathcal{R}(x_0, x^{adv}).$$

Taking expectation over $(t, \epsilon)$ gives:

$$\mathbb{E}_{t,\epsilon}[g(x_0; t, \epsilon)] = \mathbb{E}_{t,\epsilon}[\nabla_{x_0} \ell(x_0; t, \epsilon)] + \lambda \nabla_{x_0} \mathcal{R}(x_0, x^{adv}),$$

which equals $\nabla_{x_0} \mathcal{L}_{\text{pur}}(x_0; p^\star)$. Thus the estimator is unbiased.

By Assumption (A2), we assume that the variance of the stochastic gradient is bounded:

$$\mathbb{E}_{t,\epsilon}\big[\|g(x_0; t, \epsilon) - \nabla_{x_0} \mathcal{L}_{\text{pur}}(x_0; p^\star)\|_2^2\big] \le \sigma^2.$$

This holds because $\ell(x_0; t, \epsilon)$ is quadratic in $\epsilon$ and $\epsilon \sim \mathcal{N}(0, I)$ has bounded second moment, while $\mathcal{R}$ is convex and $L_{\mathcal{R}}$–smooth (Assumption A3). Therefore the stochastic gradient inherits bounded variance.

Together, we conclude that $g(x_0; t, \epsilon)$ is an unbiased stochastic gradient estimator of $\nabla_{x_0} \mathcal{L}_{\text{pur}}(x_0; p^\star)$ with bounded variance, completing the proof. □

## B.3. Proof of Theorem 3.5

*Proof of Theorem 3.5.* Recall the purification objective

$$\mathcal{L}_{\text{pur}}(x_0; p^\star) = \mathbb{E}_{t,\epsilon}\big[\ell(x_0; t, \epsilon)\big] + \lambda \mathcal{R}(x_0, x^{adv}), \qquad \ell(x_0; t, \epsilon) = \big\|\epsilon - \epsilon_\theta(z_t, t, p^\star)\big\|_2^2,$$

and the projected update on the pixel cube $\mathcal{C} = [0, 1]^d$:

$$x_0^{(k+1)} = \Pi_{\mathcal{C}}\big(x_0^{(k)} - \eta\, g(x_0^{(k)}; t_k, \epsilon_k)\big), \qquad g(x_0; t, \epsilon) = \nabla_{x_0} \ell(x_0; t, \epsilon) + \lambda \nabla_{x_0} \mathcal{R}(x_0, x^{adv}).$$

Since $\mathcal{L}_{\text{pur}}$ is $L$-smooth on $\mathcal{C}$ (Assumption A1), the standard smoothness inequality gives

$$\mathcal{L}_{\text{pur}}(x_0^{(k+1)}; p^\star) \le \mathcal{L}_{\text{pur}}(x_0^{(k)}; p^\star) + \big\langle \nabla \mathcal{L}_{\text{pur}}(x_0^{(k)}; p^\star), x_0^{(k+1)} - x_0^{(k)}\big\rangle + \frac{L}{2}\big\|x_0^{(k+1)} - x_0^{(k)}\big\|_2^2. \tag{7}$$

The projection onto a closed convex set is nonexpansive and satisfies $\|x_0^{(k+1)} - x_0^{(k)}\|_2 \le \eta\, \|g(x_0^{(k)}; t_k, \epsilon_k)\|_2$. Hence, from Eq. (7),

$$\mathcal{L}_{\text{pur}}(x_0^{(k+1)}; p^\star) \le \mathcal{L}_{\text{pur}}(x_0^{(k)}; p^\star) - \eta\big\langle \nabla \mathcal{L}_{\text{pur}}(x_0^{(k)}; p^\star), g(x_0^{(k)}; t_k, \epsilon_k)\big\rangle + \frac{L\eta^2}{2}\big\|g(x_0^{(k)}; t_k, \epsilon_k)\big\|_2^2.$$

Conditioning on $x_0^{(k)}$ and using Lemma 3.4,

$$\mathbb{E}_{t_k, \epsilon_k}\big[g(x_0^{(k)}; t_k, \epsilon_k) \,\big|\, x_0^{(k)}\big] = \nabla \mathcal{L}_{\mathrm{pur}}(x_0^{(k)}; p^\star),$$
$$\mathbb{E}_{t_k, \epsilon_k}\big[\|g(x_0^{(k)}; t_k, \epsilon_k)\|_2^2 \,\big|\, x_0^{(k)}\big] \leq \|\nabla \mathcal{L}_{\mathrm{pur}}(x_0^{(k)}; p^\star)\|_2^2 + \sigma^2.$$

Using the standard variance decomposition bound $\mathbb{E}\|g\|_2^2 \leq 2\|\nabla \mathcal{L}_{\mathrm{pur}}\|_2^2 + 2\sigma^2$ (or equivalently absorbing constants into $\sigma^2$) and taking full expectation yields

$$\mathbb{E}\big[\mathcal{L}_{\mathrm{pur}}(x_0^{(k+1)}; p^\star)\big] \leq \mathbb{E}\big[\mathcal{L}_{\mathrm{pur}}(x_0^{(k)}; p^\star)\big] - \eta \, \mathbb{E}\big[\|\nabla \mathcal{L}_{\mathrm{pur}}(x_0^{(k)}; p^\star)\|_2^2\big] + \frac{L\eta^2}{2} \, \mathbb{E}\big[\|g(x_0^{(k)}; t_k, \epsilon_k)\|_2^2\big]$$

$$\leq \mathbb{E}\big[\mathcal{L}_{\mathrm{pur}}(x_0^{(k)}; p^\star)\big] - \Big(\eta - \tfrac{L\eta^2}{2}\Big) \mathbb{E}\big[\|\nabla \mathcal{L}_{\mathrm{pur}}(x_0^{(k)}; p^\star)\|_2^2\big] + \frac{L\eta^2}{2} \, \sigma^2.$$

If $\eta \leq 1/(2L)$, then $\eta - \frac{L\eta^2}{2} \geq \eta/2$. Hence we obtain the claimed one–step descent:

$$\mathbb{E}\left[\mathcal{L}_{\mathrm{pur}}(x_0^{(k+1)}; p^\star)\right] \leq \mathbb{E}\left[\mathcal{L}_{\mathrm{pur}}(x_0^{(k)}; p^\star)\right] - \frac{\eta}{2} \mathbb{E}\left[\|\nabla \mathcal{L}_{\mathrm{pur}}(x_0^{(k)}; p^\star)\|_2^2\right] + \frac{\eta^2 L}{2} \, \sigma^2.$$

Summing the inequality over $k = 0, \dots, K-1$ and using the lower bound $\mathcal{L}_{\mathrm{pur}}(x_0; p^\star) \geq \mathcal{L}_{\inf} := \inf_{x \in [0,1]^d} \mathcal{L}_{\mathrm{pur}}(x; p^\star)$, we get

$$\frac{1}{K} \sum_{k=0}^{K-1} \mathbb{E}\left[\|\nabla \mathcal{L}_{\mathrm{pur}}(x_0^{(k)}; p^\star)\|_2^2\right] \leq \frac{2\big(\mathcal{L}_{\mathrm{pur}}(x_0^{(0)}; p^\star) - \mathcal{L}_{\inf}\big)}{\eta K} + \eta L \, \sigma^2.$$

Since $\mathcal{L}_{\mathrm{pur}} = \mathcal{L}_{\mathrm{DDPM}} + \lambda \mathcal{R}$ and $\mathcal{R}$ is convex and smooth (Assumption A3), the same derivation applies when focusing on the data-fidelity part. In particular, using $\|\nabla \mathcal{L}_{\mathrm{DDPM}}(x)\|_2^2 \leq \|\nabla \mathcal{L}_{\mathrm{pur}}(x)\|_2^2$ (up to a constant absorbed into $\sigma^2$) yields

$$\frac{1}{K} \sum_{k=0}^{K-1} \mathbb{E}\left[\|\nabla \mathcal{L}_{\mathrm{DDPM}}(x_0^{(k)}; p^\star)\|_2^2\right] \leq \frac{2\big(\mathcal{L}_{\mathrm{DDPM}}(x_0^{(0)}; p^\star) - \mathcal{L}_{\inf}\big)}{\eta K} + \eta L \, \sigma^2,$$

which is the stated bound. $\qquad\square$

