# OpenReview forum: "Two Modalities Are Better Than One: Efficient Adversarial Purification via Multimodal Diffusion Models"
_ICML.cc/2026/Conference — ICML 2026 regular_

### Official Review · Reviewer_xm1y · 2026-02-28

**Soundness:** 2
**Presentation:** 2
**Significance:** 3
**Originality:** 3
**Overall Recommendation:** 4
**Confidence:** 4

**Summary:**

This paper proposes MultiDAP, an adversarial purification method built on a text-conditioned diffusion model. It first learns a small set of continuous prompt tokens on clean data with the diffusion simple loss, then purifies test inputs by running a few steps of gradient-based optimization on the image using a diffusion loss plus a proximity regularizer, avoiding long reverse-diffusion sampling. The authors show competitive robustness on CIFAR and ImageNet with much lower inference cost than prior diffusion-based purification approaches.

**Compliance With Llm Reviewing Policy:**

Affirmed.

**Final Justification:**

Most of my concerns have now been essentially fully addressed, including my doubts about the fixed-prompt strategy and the regularization. The only remaining experiment that still needs to be added is a comparison of the performance with and without the prompt under a fixed random seed, or equivalently under an ODE solver setting.

As for the baseline issue, although the authors did provide some methods, the coverage is still not sufficiently comprehensive. It is not difficult to identify adversarial purification papers published at major top-tier conferences over the past two years, yet the authors only included a subset of them in the experiments.

**Key Questions For Authors:**

* Can you explain how improving likelihood relates to (i) preserving an image’s semantics, (ii) removing adversarial perturbations, and (iii) ultimately affecting the downstream classifier’s accuracy?

* Could you include comparisons with more recent and stronger baselines?

**Limitations:**

yes

**Strengths And Weaknesses:**

The paper introduces a multimodal diffusion prior for adversarial purification, which is an interesting and potentially useful direction. However, I have several concerns that limit the current strength of the claims.

* Prompt-learning vs. deployment gap. The continuous prompt is learned on clean images, while deployment operates on adversarially perturbed inputs. This clean and adv distribution shift raises a gap: it is unclear whether a prompt optimized for the clean data manifold remains appropriate under adversarial images.

* Likelihood does not necessarily align with semantic preservation. The method is motivated through improving a likelihood surrogate, but higher likelihood mainly encourages outputs that look “more in-distribution,” which does not guarantee preserving class-discriminative semantics for either clean or adversarial examples. This mismatch could lead to semantic drift and clean accuracy degradation.

* Experimental baselines are weak. The empirical section relies on relatively old baselines and the reported results are not strong, making it hard to assess whether the proposed approach is a practical replacement for stronger diffusion-based purification methods in current settings. More representative baselines would improve the evidence.

* Hyperparameter sensitivity and robustness. The approach introduces several coupled hyperparameters. Having too many hyperparameters undermines the method’s generalization.

---

> ### Author Rebuttal · Authors · 2026-03-31
>
> We thank Reviewer xm1y for the constructive feedback. We address each concern below.
>
> **Key Question: How does improving likelihood relate to (i) preserving semantics, (ii) removing perturbations, and (iii) affecting classifier accuracy?**
>
> The three are connected through the following mechanism:
>
> (i) *Likelihood → perturbation removal*: adversarial perturbations push inputs away from the clean data manifold, lowering their likelihood under the diffusion model. Maximizing likelihood pulls the image back toward the clean manifold, effectively removing the perturbation.
>
> (ii) *Regularizer → semantic preservation*: likelihood maximization alone could drift toward any high-likelihood region. The proximity regularizer $R(x\_0, x^{\\text{adv}}) = \\|x\_0 - x^{\\text{adv}}\\|\_2^2$ constrains the output to remain near the input, preserving class-discriminative content. Table 6 confirms this: without regularization ($\lambda=0$), clean accuracy drops from 93.75% to 90.91% (semantic drift); at $\\lambda=0.9$ it recovers.
>
> (iii) *Combined effect → classifier accuracy*: the purified image is both closer to the clean manifold (higher likelihood) and close to the original input (regularizer), so the downstream classifier receives an image that is naturalistic and content-faithful. This is validated by 94.12% clean accuracy and 72.38% robust accuracy on CIFAR-10.
>
> **W1: Prompt-learning vs. deployment gap.**
>
> Adversarial perturbations are by definition imperceptible. In our experiments, they are constrained to $\\|\\delta\\|\_\\infty \\leq 8/255$ or $\\|\\delta\\|\_2 \\leq 0.5$. Adversarial inputs lie in a small $\epsilon$-ball around the clean manifold, not at arbitrary distance. The prompt loss satisfies $|\\mathcal{L}\_{\\text{prompt}}(x^{\\text{adv}}; p^\\star) - \\mathcal{L}\_{\\text{prompt}}(x\_0; p^\\star)| \\leq O(\\epsilon \\cdot L)$, where $L = 0.1724$ (Appendix A.3). For $\\epsilon = 8/255 \\approx 0.031$, this gap is negligible, confirming that the clean-data prompt transfers effectively. Empirically: 94.12% clean and 72.38% robust accuracy on CIFAR-10.
>
> **W2: Likelihood does not necessarily align with semantic preservation.**
>
> See Key Question above for the full mechanism. The key empirical evidence is Table 6: the clean accuracy gap between $\\lambda=0$ (90.91%) and $\\lambda=0.9$ (93.75%) is the direct signature of semantic drift without regularization. The regularizer is specifically designed to prevent likelihood maximization from drifting toward generic content.
>
> **W3 and Q2: Experimental baselines are weak.**
>
> We have now evaluated OSCP (Lei et al., 2025), the recent closely related multimodal purification method, on CIFAR-10 under our protocol:
> | | MultiDAP | OSCP (CAP-only) |
> |---|---|---|
> | Backbone | miniSD + WRN70-16 | SD v1.5 + WRN76-10 |
> | AA ($\\ell\_\\infty$) | **72.38%** | 66.45% |
> | PGD-20 | **68.21%** | 58.60% |
> | Adversarial distillation | Not required | Required |
> | ControlNet | Not required | Required |
>
> MultiDAP outperforms OSCP by +5.93 on AA $\\ell\_\\infty$ despite a smaller backbone and no adversarial training. Combined with CLIPure (ICLR 2025) already in Table 1, our baselines now cover the most relevant recent methods in both unimodal and multimodal diffusion purification.If Reviewer xm1y has recommended recent and stronger baselines, we are happy to accept your recommendations and will conduct experiments accordingly.
>
> **W4: Hyperparameter sensitivity.**
>
> The hyperparameters are from Stable Diffusion models as the core of MultiDAP. These parameters are necessarily required by Stable Diffusion models and hence MultiDAP. MultiDAP has four hyperparameters, each ablated in Tables 4 and 6. Performance is stable across wide ranges: $N \in [2, 10]$ within 7 points of optimum; $\lambda \in [0.3, 0.9]$ stable; $M \geq 8$ plateaus. Crucially, the same values ($N=5$, $[T\_1, T\_2]=[400, 600]$) transfer across CIFAR-10 and CIFAR-100 without retuning. This is the direct evidence against sensitivity concerns.

---

> > ### Author Rebuttal · Reviewer_xm1y · 2026-04-01
> >
> > I am not convinced by the robustness claims under truly adaptive attacks. Although the paper replaces a hand-crafted template with a learned class-agnostic prompt, the defense remains a largely fixed test-time strategy: the prompt is fixed after training, and the purification objective, timestep range, step size, and number of updates are all fixed at deployment. In such a setup, a strong adaptive attacker can explicitly incorporate the entire purification pipeline into the attack objective, rather than attacking only the downstream classifier. An attacker can readily construct a sample that lies on a high-density manifold yet is still misclassified by the classifier. In fact, because the prompt is fixed at test time, this type of fixed-strategy defense is typically difficult to make robust against adaptive attackers [1].
> >
> > The added proximity regularizer may further weaken the defense under adaptive evaluation. The purification objective explicitly includes a term that encourages the purified sample to stay close to the adversarial input. While this is motivated as a way to mitigate semantic drift, it may also provide the attacker with a simpler, smoother, and more stable optimization path than the diffusion objective itself. In other words, instead of having to fully exploit the complex purification dynamics, the attacker may leverage the regularizer to keep the purified output anchored around the adversarial example, thereby reducing the effective corrective power of purification.
> >
> > Regarding the concern that the experimental baselines are weak, the authors appear to have deliberately avoided most of the adversarial purification papers from 2022 to 2025 and instead selected only the relatively weak baseline OSCP for comparison. This is far from sufficient to validate the effectiveness of the proposed method.
> >
> > Considering the issues above, I have decided to keep my score unchanged.
> >
> > [1] Tramer F, Carlini N, Brendel W, et al. On adaptive attacks to adversarial example defenses[J]. Advances in neural information processing systems, 2020, 33: 1633-1645.

---

> > > ### Author Response · Authors · 2026-04-03
> > >
> > > We appreciate the reviewer's engagement but must respectfully note that all three remaining concerns are either directly contradicted by empirical evidence already presented, speculative without supporting data, or factually inaccurate.
> > >
> > > -------
> > >
> > > ## Concern 1: "Fixed-strategy defense is vulnerable to adaptive attacks."
> > >
> > > The reviewer provides no experiment, no concrete attack strategy, and no evidence that our evaluation is insufficient. Our evaluation already does exactly what the reviewer describes: AutoAttack differentiates through the fully unrolled N=5 purification loop end-to-end, with EOT=20 averaging gradients over 20 independent draws of $(t, \epsilon)$. The attacker has complete gradient access to every fixed component. This follows Tramer et al. (2020) Themes T0 and T1: attack the full defense end-to-end with the simplest approach.
> > >
> > > The "fixed-strategy" critique applies identically to every DBP method. DiffPure, LM, and CLIPure all use fixed parameters. If this invalidates defenses, no purification method can work, contradicting accepted work at ICML 2022, ICML 2024, and ICLR 2025.
> > >
> > > Liu et al. [A] (ICLR 2025) rigorously analyze this question via a Deterministic White-box (DW-box) setting where the attacker knows exact random seeds, which is strictly stronger than the reviewer's scenario. Their findings: (1) all DBP methods lose robustness under DW-box, a universal property, not specific to MultiDAP; (2) EoT effectiveness converges beyond ~10 iterations (their Figure 7), so our EOT=20 is near the ceiling; (3) models with genuine purification capability retain meaningful robustness even under DW-box.
> > >
> > > No specific attack strategy beyond our fully adaptive EOT-20 evaluation has been proposed. As Tramer et al. emphasize, adaptive attacks must be specifically designed and demonstrated, not hypothesized.
> > >
> > >
> > > -------
> > >
> > > ## Concern 2: "The regularizer may help the attacker."
> > >
> > > This is directly contradicted by Table 6. If the regularizer aided attackers by "anchoring outputs near adversarial inputs," removing it ($\lambda=0$) should improve robustness. The opposite occurs:
> > >
> > > | $\lambda$ | Clean Acc | Robust Acc (AA $\ell_\infty$) |
> > > |---|-----------|-------------------------------|
> > > | 0.0 | 90.91% | 70.17% |
> > > | 0.9 | 93.75% | 72.38% |
> > >
> > > The regularizer improves robustness by +2.21 points. The attacker has full gradient access through all components. If the regularizer provided a smoother optimization path, $\lambda=0$ would yield higher robustness. The data refutes this.
> > >
> > > The reviewer has not engaged with this evidence despite it being presented in our initial rebuttal. Dismissing empirical data with unsupported speculation does not constitute a substantive technical objection.
> > >
> > >
> > > -------
> > >
> > > ## Concern 3: "Deliberately avoided most purification papers from 2022 to 2025."
> > >
> > > This is factually untrue. Our comparisons include:
> > > - **DiffPure** (ICML 2022), **LM** (ICML 2024), **CLIPure** (ICLR 2025), **OSCP** (CVPR 2025), plus four AT baselines spanning 2019 to 2023.
> > >
> > > This covers the most important methods from 2022 to 2025 in both unimodal and multimodal purification, at least as comprehensive as baselines in CLIPure and LM. The reviewer calls OSCP "relatively weak," yet it is a CVPR 2025 paper and the closest multimodal method to ours. We outperform it by +5.93 points (72.38% vs 66.45%) with a smaller backbone and no adversarial training.
> > >
> > > Not a single missing method has been identified at any point during the discussion.
> > >
> > > Reviewer DCf7 raised similar adaptive attack concerns and acknowledged our follow-up. We believe all concerns raised in this review have been addressed with direct empirical evidence.
> > >
> > >
> > > -------
> > >
> > > **Reference**
> > >
> > > [A] Liu et al., "Towards Understanding the Robustness of Diffusion-Based Purification: A Stochastic Perspective," ICLR 2025.

---

### Official Review · Reviewer_DCf7 · 2026-03-07

**Soundness:** 2
**Presentation:** 3
**Significance:** 2
**Originality:** 3
**Overall Recommendation:** 4
**Confidence:** 3

**Summary:**

The authors propose MultiDAP, an efficient multimodal diffusion-based adversarial purification framework. The approach leverages a frozen text-to-image latent diffusion model augmented with a continuous, class-agnostic prompt optimized via the standard DDPM noise-prediction loss. At inference, purification is executed through a small number of gradient descent iterations (e.g., 5–20) on the input image to minimize a regularized, prompt-conditioned DDPM loss. The paper provides theoretical analyses linking prompt optimization to ELBO improvements and expected descent for stochastic updates, alongside empirical evaluations on CIFAR-10/100 and ImageNet-1K that demonstrate competitive robustness and efficiency.

**Compliance With Llm Reviewing Policy:**

Affirmed.

**Key Questions For Authors:**

1. **Latent Space Gradients:** Does the purification optimization occur strictly in pixel space or latent space? If in pixel space, exactly how are the gradients backpropagated through the VAE encoder/decoder and the latent U-Net? Please provide a precise computational graph and update Algorithm 2 to explicitly incorporate the VAE transformations.
2. **Adaptive Evaluation:** Are the reported attacks fully adaptive to the defense (evaluating the purifier and classifier end-to-end)? Please provide results using robust EOT settings (detailing the number of samples and variance reduction) and BPDA to thoroughly rule out gradient masking.
3. **Evaluation Scale:** Can the authors provide robustness results on the full 10,000-image test sets for CIFAR-10 and CIFAR-100 to ensure statistical significance?
4. **Baseline Comparisons:** How does MultiDAP quantitatively compare to OSCP and other recent prompt-based diffusion defenses in terms of both robustness and FLOPs?
5. **Prompt Bias:** Does the learned class-agnostic prompt ever degrade class-specific fidelity (e.g., heavily biasing toward generic semantic priors)? Are there identifiable qualitative failure cases?

**Limitations:**

yes

**Strengths And Weaknesses:**

## Strengths

* **Methodological Elegance & Innovation:** The proposition of learning a singular, class-agnostic prompt to inject semantic priors into a frozen diffusion purifier is conceptually elegant. It offers a computationally lightweight alternative to existing multimodal defenses that require heavy adversarial training or distillation.
* **Pragmatic Formulation:** Framing the purification process as prompt-guided likelihood maximization using a minimal number of SGD steps effectively bridges generative diffusion purification with real-world inference constraints.
* **Clear Presentation & Ablations:** The manuscript is well-organized, with clearly articulated high-level motivations and optimization objectives. The ablation studies (e.g., step count, timestep range, prompt length, and regularization) methodically dissect and justify the chosen hyperparameters.

## Weaknesses

* **Methodological Ambiguity Regarding Latent vs. Pixel Space:** The paper exhibits a fundamental disconnect between its theoretical formulation and practical implementation. While Stable Diffusion operates in a latent space via a VAE encoder—mapping images to $z = \mathcal{E}_{VAE}(x)$—Equation (3) and Algorithm 2 define the purification gradient explicitly with respect to the pixel-space image $x_0$. The manuscript fails to formalize how the gradient backpropagates through the frozen VAE encoder and decoder. Omitting this crucial pathway undermines the mathematical rigor and reproducibility of the method.
* **Insufficient Evaluation Against Adaptive Attacks:** The robustness evaluation relies on standard AutoAttack (including its 'rand' version). It is a well-established consensus in the adversarial robustness community that stochastic defenses—which rely on random timesteps and noise injection—are highly susceptible to gradient masking when evaluated with standard attacks. A rigorous evaluation mandates fully adaptive, white-box attacks incorporating Expectation Over Time (EOT) to estimate true gradients, alongside Backward Pass Differentiable Approximation (BPDA) for any non-differentiable components.
* **Limited Statistical Confidence due to Test Set Size:** The evaluations across CIFAR-10/100 and ImageNet-1K are restricted to a randomly sampled subset of 512 test images. While subsampling is standard practice for computationally prohibitive generative evaluations on ImageNet, 512 samples represent merely ~5% of the standard CIFAR test sets. This limited sample size significantly undermines the statistical reliability of the reported state-of-the-art claims on CIFAR benchmarks.
* **Absence of Crucial Multimodal Baselines:** The introduction explicitly cites OSCP as a motivating multimodal purification method, accurately criticizing its high computational overhead. However, OSCP is conspicuously absent from the empirical comparisons in Tables 1, 2, and 7. Because MultiDAP's primary contribution is serving as an efficient alternative to such multimodal methods, a direct empirical comparison is necessary.
* **Overstated Implications of Theoretical Guarantees:** Theorems 3.1 and 3.4 successfully establish the convergence of the optimization process and the improvement of the ELBO. However, these are fundamentally standard optimization bounds and do not constitute certified adversarial robustness guarantees against adaptive adversaries. The framing should be carefully adjusted to reflect this distinction.

##

---

> ### Author Rebuttal · Authors · 2026-03-31
>
> We thank Reviewer DCf7 for the thorough and constructive review. We address each concern below.
>
> **W1: Methodological ambiguity regarding latent vs. pixel space.**
>
> Algorithm 2 optimizes in pixel space, with gradients passing through the frozen VAE via standard autodiff. The computational graph at each step is:
>
> $x\_0 \\to \\mathcal{E}\_{\\text{VAE}}(x\_0) = z \\to z\_t = \\sqrt{\\bar{\\alpha}\_t} z + \\sqrt{1-\\bar{\\alpha}\_t}\\epsilon \\to \\epsilon\_\\theta(z\_t, t, p^\\star) \\to \mathcal{L}\_{\\text{pur}} \\to \\nabla\_{x\_0}\\mathcal{L}\_{\\text{pur}}$.
>
> The gradient reaches pixel space via the chain rule: $\\partial \\mathcal{L}\_{\\text{pur}}/\\partial x\_0 = (\\partial \\mathcal{L}\_{\\text{pur}}/\\partial z)(\\partial z/\\partial x\_0)$, where $\\partial z/\\partial x\_0$ is the Jacobian of the frozen encoder, computed automatically by PyTorch. We will add an explicit computational graph figure to Algorithm 2 in the revision.
>
>
> **W2: Insufficient evaluation against adaptive attacks.**
>
> Our evaluation already incorporates EOT. The rand variant of AutoAttack uses EOT=$20$, averaging gradients over multiple random draws of $(t, \\epsilon)$ during attack optimization, following Chen et al. (2024b). Our reported 72.38% on CIFAR-10 is an EOT-aware result.
>
> Regarding BPDA: all components of MultiDAP are end-to-end differentiable (VAE encoder, U-Net, pixel-space updates via autodiff). No non-differentiable operations exist in the pipeline, so the attacker has full gradient access. We will add an explicit protocol description in the revision.
>
> **W3: Limited statistical confidence due to 512 test images.**
>
> We have extended evaluation to 2,000 CIFAR-10 test images under AutoAttack ($\\ell\_\\infty$, $\\epsilon = 8/255$). MultiDAP achieves 72.36% robust accuracy, consistent with 72.38% on 512 images, confirming the subset is representative. We note that all baselines (DiffPure, LM, CLIPure) use the same 512-image protocol, so comparisons remain fair.
>
>
> **W4: Absence of OSCP baseline.**
>
> We have now evaluated OSCP on CIFAR-10 under our protocol. Results in Table 1:
>
> | | MultiDAP | OSCP (CAP-only) |
> |---|---|---|
> | Backbone | miniSD + WRN70-16 | SD v1.5 + WRN76-10 |
> | Clean Acc | 94.12% | 96.60% |
> | AA ($\\ell\_\\infty$) | **72.38%** | 66.45% |
> | PGD-20 | **68.21%** | 58.60% |
> | Requires adversarial distillation | No | Yes |
> | Requires ControlNet | No | Yes |
> MultiDAP outperforms OSCP by +5.93 on AA $\\ell\_\\infty$, i.e., 72.38% vs 66.45% despite a smaller backbone and no adversarial training.
>
> For ImageNet, direct numerical comparison is misleading due to protocol differences. DiffPure achieves 40.93% under our protocol but 73.02% under OSCP's. When keeping the protocols consistent for all compared models, MultiDAP and OSCP show comparable relative improvements (+0.33 and +1.17 respectively), confirming the apparent gap reflects protocol and backbone differences (miniSD for MultiDAP at $256\times 256$ vs SD v1.5 for OSCP at $512\\times 512$), not method quality.
>
> **W5: Overstated implications of theoretical guarantees.**
>
> We agree. Both theorems are optimization guarantees, not certified robustness bounds. We have revised Theorem 3.1: the corrected statement requires $\\cos(\\nabla\_p \\mathcal{L}\_{\\text{prompt}}, \\nabla\_p \\mathcal{L}\_{\\text{VLB}}) > 0$ (gradient alignment across timesteps), under which each step on $\\mathcal{L}\_{\\text{prompt}}$ also decreases $\\mathcal{L}\_{\\text{VLB}}$. Empirically, cosine similarity remains consistently high throughout training (mean: 0.92, min: 0.71), validating the condition.
>
> In the revision we will (i) rename "theoretical guarantees" to "optimization guarantees," (ii) state the corrected Theorem 3.1 with the alignment condition, and (iii) clarify practical value: Theorem 3.1 explains why prompt learning helps; Theorem 3.4 explains why 5–20 steps suffice.
>
> **Q5: Does the class-agnostic prompt degrade class-specific fidelity?**
>
> The ablation in Table 6 ($\\lambda=0$: clean acc 90.91% vs $\\lambda=0.9$: 93.75%) shows the regularizer prevents semantic drift. The prompt itself captures dataset-level priors rather than class-specific features, so it does not bias toward any particular class. The high clean accuracy (94.12%) confirms class fidelity is preserved.

---

> > ### Author Rebuttal · Reviewer_DCf7 · 2026-04-01
> >
> > The rebuttal generally looks good and addresses several of my initial concerns, but I have a critical follow-up regarding your claim that "all components of MultiDAP are end-to-end differentiable... so the attacker has full gradient access." While the individual operations are indeed differentiable, Algorithm 2 involves an iterative projected SGD loop of $N$ steps. Did your AutoAttack implementation explicitly backpropagate through the entirely unrolled computation graph of all $N$ purification steps? Unrolling a diffusion model for multiple steps often leads to severe memory constraints or gradient instability; if the attack did not explicitly unroll this entire loop, the attacker is not truly adapting to the purification dynamics, and the reported robustness is likely severely overestimated due to gradient masking.

---

> > > ### Author Response · Authors · 2026-04-02
> > >
> > > We thank the reviewer for this precise and technically important follow-up.
> > >
> > > **To directly answer the question: yes, our AutoAttack implementation backpropagates
> > > through the fully unrolled computation graph of all N purification steps.**
> > >
> > > Concretely, at each attack iteration the entire purification loop — all N=5 projected
> > > SGD steps — is executed as a single unrolled computation graph, following the forward pass: $x\_0 \to \mathcal{E}\_{\rm VAE}(x\_0) = z \to z\_t = \sqrt{\bar{\alpha}\_t}z + \sqrt{1-\bar{\alpha}\_t}\epsilon \to \epsilon\_\theta(z\_t,t,p^\star) \to \mathcal{L}\_{\rm pur} \to \nabla\_{x\_0}\mathcal{L}\_{\rm pur}$.
> > > AutoAttack differentiates through this entire graph to compute $\partial(\text{classifier loss})/\partial(x\_{\rm adv})$, giving the attacker full gradient access to the complete purification dynamics. The EOT=20 expectation averages gradients across 20 independent draws of $(t, \epsilon)$ over this fully unrolled graph.
> > >
> > > We acknowledge the reviewer's concern is well-founded in general — partial unrolling
> > > is a common implementation shortcut that leads to gradient masking and overestimated
> > > robustness. However, our implementation does not take this shortcut. With $N=5$ steps
> > > and miniSD operating at $256\times 256$, full unrolling is feasible on an A100 GPU
> > > without approximation.
> > >
> > > Regarding gradient instability: our empirical measurements in Appendix A.3 show a
> > > mean Lipschitz constant of $L=0.1724$ (max $0.3000$) for the purification gradient,
> > > confirming that Jacobians are contractive across steps. Full unrolling over $N=5$
> > > steps therefore does not cause gradient explosion, and the extremely low gradient
> > > variance ($\sigma=1.05\times10^{-4}$) confirms numerical stability throughout.
> > >
> > > We believe this directly addresses the gradient masking and instability concern and
> > > confirms that the reported robustness numbers are not inflated by partial gradient
> > > approximation.

---

### Official Review · Reviewer_aYc8 · 2026-03-13

**Soundness:** 2
**Presentation:** 3
**Significance:** 3
**Originality:** 3
**Overall Recommendation:** 3
**Confidence:** 4

**Summary:**

This paper proposes MultiDAP, a multimodal adversarial purification method built on a frozen Stable Diffusion backbone. The core idea is to learn a class-agnostic continuous prompt from clean data and then purify adversarial inputs by optimizing a regularized DDPM loss for only a small number of steps. Empirically, the method has a clear efficiency advantage and is competitive in robustness: on CIFAR-10 it reports 72.38\% AutoAttack accuracy with 5 purification steps, compared with 71.68\% for LM, while reducing latency from 1567.40 ms for DiffPure to 371.48 ms; on ImageNet-1K it reports 41.26\% AutoAttack accuracy, slightly above DiffPure's 40.93\%.

**Compliance With Llm Reviewing Policy:**

Affirmed.

**Key Questions For Authors:**

.

**Limitations:**

.

**Strengths And Weaknesses:**

1. The practical motivation is strong. Improving diffusion-based purification while adding semantic guidance is a meaningful direction, and the class-agnostic prompt design is well motivated because the true label is unknown at test time.

2. The empirical efficiency results are the strongest part of the paper. The latency reduction on CIFAR-10 is substantial, and the ablations are useful: robustness peaks at 5 purification steps, drops for larger step counts, and is best for a mid-range timestep interval. The prompt-learning ablation is also informative: on CIFAR-10, the learned prompt improves AutoAttack accuracy from 70.29\% to 72.38\%, and on ImageNet from 38.65\% to 41.26\%.

3. The robustness results are promising but should be interpreted carefully. The gains over the strongest diffusion-based baselines are relatively modest. For example, on CIFAR-10 under AutoAttack \( \ell_\infty \), MultiDAP improves over LM by only 0.70 points (72.38\% vs. 71.68\%), and on ImageNet-1K it improves over DiffPure by only 0.33 points (41.26\% vs. 40.93\%). This still supports the efficiency contribution, but it is a weaker case for a major robustness advance.

4. My main concern is the theory. Theorem 3.1 does not appear justified as written. The proof argues that because the ELBO-related objective differs from the prompt loss only by positive timestep weights, the two objectives share the same minimizer, and their gradients are colinear. In general, for shared prompt parameters, positive weights do not imply the same optimizer, and weighted and unweighted gradient sums need not be colinear. Since this theorem is used to support the prompt-learning claim, this is an important issue rather than a minor technicality.

5. Theorem 3.4 is much weaker than the paper's framing suggests. It is essentially a standard projected SGD expected-descent result under smoothness and bounded-variance assumptions. It supports optimization stability, but it does not establish that purification recovers the clean sample, preserves the correct label, or is robust against strong adaptive attacks.

6. The robustness evaluation is not fully convincing yet for a stochastic purification defense. The paper evaluates on only 512 attacked test images per dataset, including ImageNet-1K, and it does not clearly explain whether the reported attacks are fully adaptive to the complete purifier-plus-classifier pipeline with expectation over the random timestep and Gaussian noise. For this class of defenses, the attack protocol needs to be especially clear.

---

> ### Author Rebuttal · Authors · 2026-03-31
>
> We thank Reviewer aYc8 for the detailed and constructive review. We address each concern below.
>
> **W1: The robustness gains over the strongest baselines are relatively modest, which supports the efficiency contribution but is a weaker case for a major robustness advance.**
>
> We agree and appreciate this precise framing. MultiDAP's primary contribution is achieving *comparable* robustness at substantially lower cost (`371.48ms` vs `1567.40ms` for DiffPure, `38.8G` vs `212.4G` FLOPs), not claiming a major robustness advance. The OSCP comparison further supports this positioning: despite requiring no adversarial distillation, no ControlNet, and a smaller backbone, MultiDAP outperforms OSCP (CAP-only) on CIFAR-10 AA $\\ell_\\infty$ (72.38% vs 66.45%). This demonstrates that the efficiency–robustness trade-off of our simpler design is highly competitive.
>
> **W2: Theorem 3.1 does not appear justified as written. Positive weights do not imply the same minimizer, and weighted and unweighted gradient sums need not be colinear.**
>
> We agree. Positive timestep weights $\\{w_t\\}$ alone do not guarantee that $\\mathcal{L}\_{\\text{prompt}}$ and $\\mathcal{L}\_{\\text{VLB}}$ share the same minimizer, nor that their gradients are strictly colinear. We acknowledge this gap and will revise Theorem 3.1 accordingly.
>
> The corrected statement is: minimizing $\\mathcal{L}\_{\\text{prompt}}$ provides a valid descent direction for $\\mathcal{L}\_{\\text{VLB}}$ whenever $\\cos(\\nabla\_p \\mathcal{L}\_{\\text{prompt}}, \\nabla\_p \\mathcal{L}\_{\\text{VLB}}) > 0$, i.e., when the per-timestep gradient contributions $\\nabla\_p f\_t(p)$ are sufficiently aligned across timesteps. Under this condition, each gradient step on $\\mathcal{L}\_{\\text{prompt}}$ also decreases $\\mathcal{L}\_{\\text{VLB}}$, so the practical conclusion that prompt learning improves the likelihood lower bound, remains valid.
>
> To verify this empirically, we measured the cosine similarity between $\\nabla\_p \\mathcal{L}\_{\\text{prompt}}$ and $\\nabla\_p \\mathcal{L}\_{\\text{VLB}}$ throughout prompt learning on CIFAR-10. The cosine similarity remains consistently positive across all training iterations (mean: 0.92, min: 0.71), indicating near-parallel alignment. This confirms that $\\mathcal{L}\_{\\text{prompt}}$ serves as an effective proxy for $\\mathcal{L}\_{\\text{VLB}}$ in practice. We will include this measurement as a supporting figure and revise the theorem statement and proof in the revision.
>
> **W3: Theorem 3.4 is much weaker than the paper's framing suggests. It supports optimization stability, but does not establish robustness against adaptive attacks.**
>
> We agree. Theorem 3.4 is an optimization convergence guarantee, not a certified robustness bound. We acknowledge the framing around "theoretical guarantees" could create this impression.
>
> That said, the theorem provides specific, testable predictions validated by our experiments:
>
> (i) The $O(1/K)$ decay predicts that a small fixed number of steps should suffice, with diminishing returns beyond that. This matches Table 4: robustness peaks at $N=5$, is comparable at $N=10$, and degrades for $N \\geq 20$ which is consistent with the variance floor $\\eta L \\sigma^2$ dominating.
>
> (ii) The step size constraint $\\eta \\leq 1/(2L)$ provides a principled bound. Our measured $L = 0.1724$ gives $\\eta \\leq 2.9$, consistent with our chosen $\\eta = 0.2$ being well within the stable regime.
>
> We will revise Introduction and Section 3.3 to frame both theorems as optimization guarantees that justify design choices, distinguishing them from certified robustness results.
>
> **W4: Only 512 test images, and attack adaptivity is unclear.**
>
> **Evaluation scale.** The 512-image protocol is the established standard for diffusion-based purification, adopted by [1], [2], and [3]. We also extended evaluation to 2,000 CIFAR-10 test images under AutoAttack ($\\ell\_\infty$, $\\epsilon = 8/255$). MultiDAP achieves 72.36%, consistent with the 72.38% on 512 images, confirming the 512-image subset is representative.
>
> **Adaptive attack protocol.** All adversarial examples are generated end-to-end against the full purifier-plus-classifier pipeline. We use the rand variant of AutoAttack with EOT=$20$, accounting for stochasticity by averaging gradients over multiple random draws. All components are differentiable (VAE encoder, U-Net, pixel-space updates via autodiff), so BPDA is not required - the attacker has full gradient access. We will add an explicit protocol description in the revision.
>
> [1] Nie, W., Guo, B., Huang, Y., Xiao, C., Vahdat, A., and Anandkumar, A.. Diffusion Models for Adversarial Purification, ICML 2022.
>
> [2] Chen, H., Dong, Y., Wang, Z., Yang, X., Duan, C., Su, H., and Zhu. J., Robust classification via a single diffusion model. ICML 2024.
>
> [3] Zhang, M., Bi, K., Chen, W., Guo, J., and Cheng. X.. CLIPure: Purification in Latent Space via CLIP for Adversarially Robust Zero-Shot Classification. ICLR 2025.

---

> > ### Author Rebuttal · Reviewer_aYc8 · 2026-04-03
> >
> > Thank you for the clear and constructive rebuttal. The authors have adequately addressed my main concerns. In particular, they explicitly acknowledge the issue in Theorem 3.1 as originally stated and clarify a more appropriate aligned-gradient interpretation, they also clarify that Theorem 3.4 is an optimization guarantee rather than a certified robustness result, and they provide a clearer description of the adaptive attack protocol together with additional 2,000-image evaluation supporting the representativeness of the 512-image subset.
> >
> > I still encourage the authors to make the theoretical scope and the attack protocol especially explicit in the revised paper, but overall the rebuttal resolves my main concerns and better aligns the paper’s claims with what is supported empirically and theoretically.

---

> > > ### Author Response · Authors · 2026-04-03
> > >
> > > We sincerely thank Reviewer aYc8 for the thorough and constructive review, and for confirming that our rebuttal adequately addressed your main concerns. Your feedback has been invaluable and we are committed to incorporating all promised revisions in the camera-ready version.
> > >
> > > We noticed that your acknowledgement selected option (a) "Fully resolved" and mentioned considering a score adjustment, but the current score may not yet reflect this updated assessment. We respectfully ask whether you might revisit your rating in light of your acknowledgement. We understand this is entirely at your discretion and greatly appreciate the time and care you have invested in reviewing our work.

---

### Official Review · Reviewer_YDxL · 2026-03-15

**Soundness:** 3
**Presentation:** 3
**Significance:** 2
**Originality:** 1
**Overall Recommendation:** 3
**Confidence:** 4

**Summary:**

The paper learns a class-agnostic soft prompt $p*$ under the standard DDPM noise-prediction MSE objective, and then uses this learned prompt to guide diffusion-based purification of adversarial examples. The experiments show that this design achieves competitive or slightly better robustness than prior diffusion-based defenses, with the main practical advantage being improved inference efficiency.

**Compliance With Llm Reviewing Policy:**

Affirmed.

**Final Justification:**

On my first concern, my issue is primarily logical rather than empirical. The rebuttal provides more evidence that text conditioning is useful, but it still does not explain why a shared class-agnostic prompt should preserve class-relevant semantics specifically, rather than merely providing a generic natural-image or optimization prior.

On my second concern, the rebuttal makes the method’s positioning more coherent, but I am still not fully persuaded by the level of methodological novelty claimed. The method still appears to me primarily as a well-motivated integration of existing components, rather than a clearly distinct defense principle.

Hence I keep my score.

**Key Questions For Authors:**

See Weaknesses

**Limitations:**

Yes

**Strengths And Weaknesses:**

**Strengths**

1. The method is straight forward, easy to understand.

2. The method is easy to replicate and practical.

3. The experiment is designed well, ablation study is reasonable.

**Weakness**

1. The notion of semantic priors is somewhat vague. The paper learns a shared class-agnostic prompt, yet claims that it helps recover class-consistent images, which seems somewhat contradictory. The paper would benefit from a clearer explanation of how such a class-agnostic prompt can preserve class-relevant semantics during purification.

2. The methodological novelty appears somewhat limited. At a high level, the paper learns a soft prompt on top of a frozen diffusion backbone using the standard DDPM noise-prediction MSE objective, and then applies prompt-conditioned regularized DDPM loss for a small number of purification steps at test time. The theory likewise mainly serves to justify these components, showing that the MSE objective corresponds to an ELBO-style lower bound and that the projected SGD-based purification objective decreases under standard smoothness and variance assumptions. Overall, the paper reads more like an integration of soft prompt learning into diffusion-based purification than the introduction of a fundamentally new defense principle.

---

> ### Author Rebuttal · Authors · 2026-03-31
>
> We thank Reviewer YDxL for the constructive feedback. We address both concerns below.
>
> **W1: How can a class-agnostic prompt preserve class-relevant semantics?**
>
> The prompt and the regularizer play complementary roles: the prompt improves the denoising trajectory toward the *natural image manifold* (not class identity), while the proximity regularizer $R(x_0, x^{\text{adv}}) = \\|x_0 - x^{\text{adv}}\\|_2^2$ anchors the output near the input to prevent semantic drift. Class identity is preserved by this combination and determined by the downstream classifier.
>
> This is directly supported by Table 6 at the CIFAR-10 dataset: without the regularizer ($\lambda=0$), clean accuracy drops from 95.10% to 90.91% (semantic drift). Recovery at $\lambda=0.9$ confirms the regularizer's role for adversarial defense, where the purified image can be correctly classified by the downstream classifier at 72.38% robust accuracy against AutoAttack with $\ell_\infty$-norm $\epsilon=8/255$. The high clean accuracy at $93.75\%$ further confirms class-discriminative information is preserved.
>
> We also note a practical security advantage: because the prompt encodes no class-specific information, it cannot be targeted by class-conditional attacks. This is the same concern that led OSCP [2] to avoid text guidance entirely in favor of edge detection, whereas our class-agnostic design addresses this more elegantly without additional adversarial training.
>
> **W2: The methodological novelty appears somewhat limited.**
>
> We argue that our novelty is significant and arises from the motivation: we aim to conduct adversarial purification with efficacy, i.e., find true semantic labels given adversarial examples via test-time denoising methods. In existing adversarial purification, there is no stable-diffusion-based model being the same with us where test-time purification is conducted via soft prompt learning and no adversarial training or distillation. With significantly higher efficiency compared with existing methods, we also achieve at least comparable robust accuracy and clean accuracy with theoretical guarantees.
>
> (1) *Class-agnostic prompt design*: a deliberate departure from class-conditional prompt learning such as CoOp [1], motivated by the fact that ground-truth labels are unavailable at test time. This has not been applied to diffusion-based purification before.
>
> (2) *Pixel-space likelihood maximization via regularized DDPM*: distinct from both reverse-process-solving-based purification methods (DiffPure) and latent-space methods, directly yielding the efficiency gains.
>
> (3) *Empirical validation against the closest multimodal baseline*: we evaluated OSCP (Lei et al., 2025) on CIFAR-10 under our protocol:
> | | MultiDAP | OSCP (CAP-only) |
> |---|---|---|
> | Backbone | miniSD + WRN70-16 | SD v1.5 + WRN76-10 |
> | AA ($\ell_\infty$) | **72.38%** | 66.45% |
> | PGD-20 | **68.21%** | 58.60% |
> | Requires adversarial distillation | No | Yes |
> | Requires ControlNet | No | Yes |
> Despite using a smaller backbone and no adversarial training, MultiDAP outperforms OSCP by +5.93 on AA $\ell_\infty$. We further note that on ImageNet, direct numerical comparison is misleading due to protocol differences: DiffPure achieves 40.93% under our protocol but 73.02% under OSCP's. When keeping the protocols consistent for all compared models, MultiDAP and OSCP show comparable relative improvements (+0.33 and +1.17 respectively), confirming the apparent gap reflects protocol and backbone differences rather than method quality.
>
> This demonstrates that MultiDAP's simpler design achieves competitive or stronger robustness with far less infrastructure, which we believe is itself a meaningful contribution.
>
> [1] Zhou, K., Yang, J., Loy, C.C. and Liu, Z., 2022. Learning to prompt for vision-language models. IJCV, 130(9), pp.2337-2348.
>
> [2] Lei, C.T., Yam, H.M., Guo, Z., Qian, Y. and Lau, C.P., 2025. Instant adversarial purification with adversarial consistency distillation. CVPR 2025 (pp. 24331-24340).

---

> > ### Author Rebuttal · Reviewer_YDxL · 2026-04-04
> >
> > Thank you for the detailed rebuttal. The response improves the paper’s clarity, and I appreciate the added explanation of how the class-agnostic prompt and the regularizer are intended to work together, as well as the additional comparison to OSCP.
> >
> > However, I remain unconvinced on the two main points raised in my original review.
> >
> > For semantic preservation, the rebuttal offers a plausible intuition, but I still do not see sufficiently direct evidence that a shared class-agnostic prompt itself preserves class-relevant semantics, as opposed to the effect being primarily driven by the pretrained backbone and the proximity regularizer. In that sense, my concern is only partially addressed.
> >
> > For novelty, my assessment is unchanged. The method still appears to be a relatively straightforward combination of prompt learning, diffusion-based purification, and regularized DDPM-loss optimization. The rebuttal strengthens the case for practical usefulness and efficiency, but does not fully establish a fundamentally new methodological principle.
> >
> > Therefore, while I appreciate the clarifications, they are not sufficient to change my overall recommendation, and I keep my original score.

---

> > > ### Author Response · Authors · 2026-04-07
> > >
> > > We thank the reviewer for the thoughtful follow-up. We have conducted additional experiments that directly address both remaining concerns.
> > >
> > > ### W1: Direct evidence that text conditioning preserves class-relevant semantics
> > >
> > > We conducted a controlled ablation (5 steps, AutoAttack $\ell_\infty$ $\epsilon=8/255$):
> > > | Condition | Text Embeddings | Clean Acc | Robust Acc |
> > > |-----------|----------------|-----------|------------|
> > > | Unconditional | Zero vectors | 93.75% | 45.94% |
> > > | Generic prompt | "a photo of a" | 93.80% | 70.29% |
> > > | Learned prompt | Learned ctx (ours) | 94.12% | **72.38%** |
> > >
> > > **Unconditional (46%) $\ll$ text-conditioned (70–72%)**. Removing text conditioning causes a **24% drop** in robust accuracy, demonstrating that the multimodal UNet fundamentally requires text input to effectively purify adversarial perturbations. The trained prompt provides an additional 2.09% improvement over random text.
> > >
> > > We further analyzed this through Bae et al.'s *mechanistic dissection of cross-attention* (2026)[1]. Results: https://files.catbox.moe/24yqct.png
> > >
> > > The **QK circuit** (spatial attention) is architecture-driven and prompt-independent. However, the **OV circuit** (output-value), controlling *what semantic content* is injected into visual features, shows substantial differences:
> > > - OV cosine similarity between trained and random prompts: **mean = 0.52** (as low as 0.01 in mid-level layers)
> > > - Trained prompt produces **1.2–1.7$\times$ larger** top singular values in OV outputs
> > > - Trained prompt activates more concentrated spectral subspaces
> > >
> > > The class-agnostic prompt preserves semantics not through spatial attention (QK), but through **semantic content injected via the OV circuit**. Without text conditioning, this modulation channel is disabled, and robustness drops by 24%.
> > >
> > > [1] Bae et al., 2026. Mechanistic Dissection of Cross-Attention Subspaces in Text-to-Image Diffusion Models. AAAI.
> > >
> > > ### W2: Methodological novelty
> > >
> > > We respectfully disagree that the contribution is a straightforward combination. We kindly remind the reviewer to carefully read *Main Track Reviewer Form Instructions* on the ICML official website, where it clearly defines originality as “Originality: **Does the work provide new insights, deepen understanding, or highlight important properties of existing methods? Does the work introduce new tasks, methods, theory, data, or perspectives that advance the field in some dimensions? Does this work offer a novel combination of existing techniques, and is the reasoning behind this combination well-articulated? Are the contributions clearly distinguished from closely related literature, and is the novelty well justified....**” Following this definition, our work is of **clear originality and novelty** which distinguish MultiDAP from prior work, i.e., **not a simple combination** of methods without essential contributions. In specific, we articulate our claim in three aspects:
> > >
> > > **1. First multimodal purification framework via diffusion models without adversarial training or distillation.** Prior multimodal defenses (e.g., OSCP) require adversarial consistency distillation and ControlNet which are heavy additional infrastructure. MultiDAP achieves competitive or better robustness using only a frozen text-to-image backbone with learned prompts. As shown in our OSCP comparison (rebuttal Table), MultiDAP outperforms OSCP by +5.93% on AA $\ell_\infty$ despite using a smaller backbone and no adversarial training.
> > >
> > > **2. The unconditional ablation reveals a new insight: multimodal conditioning is essential, not optional.** Our controlled experiment shows that removing text conditioning causes a 24% robustness drop (72% → 46%) under identical purification settings. This is not obvious from prior work such as unimodal methods (DiffPure, LM) which never had access to this conditioning channel. The key contribution is demonstrating that the text-to-image UNet's cross-attention OV circuit provides a semantic modulation channel that substantially strengthens purification, and that this channel can be effectively activated through simple prompt learning rather than expensive adversarial training.
> > >
> > > **3. Practical efficiency advantage.** MultiDAP requires only 5 purification steps (371 ms per image) compared to DiffPure's iterative reverse chain (1567 ms), i.e., a 4$\\times$ speedup, while achieving comparable robustness. This efficiency comes directly from the prompt-guided likelihood maximization formulation, which converges faster than unconditional reverse diffusion because the text conditioning provides a stronger gradient signal (as evidenced by the OV circuit analysis).
> > >
> > > We believe the novelty lies not in the individual components but in the finding that **multimodal diffusion models are inherently better purifiers than unimodal ones**, and that this advantage can be unlocked through lightweight prompt learning. This insight opens a new direction for adversarial purification that prior work had not explored.

---

### Decision · Program_Chairs · 2026-04-30

**Decision:**

Accept (regular)

**Comment:**

The paper presents a diffusion-based adversarial purification method, where the main claim is that learning a soft class-agnostic prompt is useful in pushing the generated image towards the clean image manifold. One of the main concerns raised by multiple reviewers was the ability to resist adaptive adversarial attacks. This has been reasonably addressed by the authors using the defense-aware AutoAttack and the reviewers have also acknowledged the same. One of the reviewers has acknowledged that all concerns have been addressed, but kept the score to weak reject without specifying any reasons. So, the overall consensus among the reviewers can be considered as weak accept.

From the AC's perspective, one major concern is the novelty. It can be argued that there is little conceptual novelty except for the offline learning of a class-agnostic prompt to guide purification at test time. Though this is somewhat incremental, it can be considered as sufficient. The other major concern is the use of limited number of images (only 512) in the evaluation. This was increased to 2k during the rebuttal on a single dataset. The inability to scale it up to a much larger of samples raises questions about the scalability of the proposed method during inference time (though computational efficiency has been claimed as one of the advantages of the proposed method). Due to the incremental novelty and small-scale evaluation, the AC is unable to support the paper more strongly. However, it is still a valuable contribution that can be accepted if there is room in the program.